# Loss Landscape Dependent Self-Adjusting Learning Rates in Decentralized Stochastic Gradient Descent

## Abstract

Distributed Deep Learning (DDL) is essential for large-scale Deep Learning (DL) training. Synchronous Stochastic Gradient Descent (SSGD) [1] is the de facto DDL optimization method. Using a sufficiently large batch size is critical to achieving DDL runtime speedup. In a large batch setting, the learning rate must be increased to compensate for the reduced number of parameter updates. However, a large learning rate may harm convergence in SSGD and training can easily diverge. Recently, Decentralized Parallel SGD (DPSGD) has been proposed to improve distributed training speed. In this paper, we find that DPSGD not only has a runtime benefit, but also a significant convergence benefit over SSGD in the large batch setting. Based on a detailed analysis of DPSGD learning dynamics, we find that DPSGD introduces additional landscape-dependent noise that automatically adjusts the effective learning rate to improve convergence. In addition, we theoretically show that this noise smooths the loss landscape, hence allowing a larger learning rate. This result also implies that DPSGD can greatly simplify learning rate tuning for tasks that require careful learning rate warmup (e.g, Attention-Based Language Modeling). We conduct extensive studies over 18 state-of-the-art DL models/tasks and demonstrate that DPSGD often converges in cases where SSGD diverges when training is sensitive to large learning rates. Our findings are consistent across three different application domains: Computer Vision (CIFAR10 and ImageNet-1K), Automatic Speech Recognition (SWB300 and SWB2000) and Natural Language Processing (Wikitext-103); three different types of neural network models: Convolutional Neural Networks, Long Short-Term Memory Recurrent Neural Networks and Attention-based Transformer Models; and two optimizers: SGD and Adam.

## 1 Introduction

Deep Learning (DL) has revolutionized AI across application domains: Computer Vision (CV) [29, 14], Natural Language Processing (NLP) [50], and Automatic Speech Recognition (ASR) [15]. Stochastic Gradient Descent (SGD) is the fundamental optimization method used in DL training. Due to massive computational requirements, Distributed Deep Learning (DDL) is the preferred mechanism to train large scale Deep Learning (DL) tasks.

The degree of parallelism in a DDL system is dictated by batch size: the larger the batch size, the more parallelism and higher speedup can be expected. However, large batches require a larger learning rate and overall they may negatively affect model accuracy because (1) large batch training usually converges to sharp minima which do not generalize well [24], and (2) large learning rates may violate the conditions (i.e., the learning rate should be less than the reciprocal of the smoothness parameter) required for convergence in nonconvex optimization theory [11]. Although training longer with large batches can lead to better generalization [18], doing so gives up some or all of the speedup we seek.

---

[1]In the literature, SSGD is also called "Centralized Synchronized Stochastic Gradient Descent". In this paper, we use these two terms interchangeably.

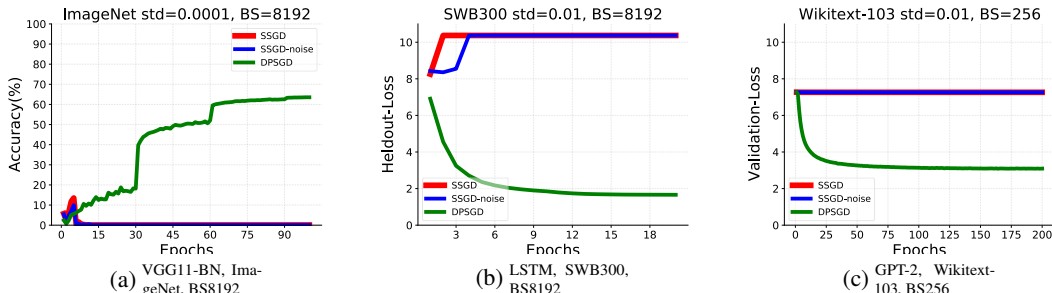

Figure 1: SSGD (red) does not converge when the learning rate needs to be large (e.g., large batch setting or a short warmup period). Figure 1a shows model accuracy (higher is better), while Figure 1b and Figure 1c show heldout loss (lower is better). Injecting Gaussian noise (blue) does not enable SSGD to escape poor local minima. In contrast, DPSGD (green) converges using the same hyperparameter setup. The detailed task descriptions and training recipes are given in Sections 4.3 and 4.5. BS denotes Batch-Size.

Through meticulous hyper-parameter design (e.g., learning rate schedules) tailored to each specific task, SSGD-based DDL systems have enabled large batch training and shortened training time for some challenging CV tasks [12, 54] and NLP tasks [55] from weeks to hours or less. However, it is observed that SSGD with large batch size leads to large training loss and inferior model quality for ASR tasks [58], as illustrated in Figure 1b (red curve). Here, we found for other types of tasks (e.g. CV and NLP) and DL models, large batch SSGD has the same problem (Figures 1a and 1c).

Several SSGD variants have been proposed to address large batch training problems: (1) local SGD, i.e., SGD-based algorithms with periodic averaging, where learners conduct global averaging after multiple steps of gradient-based updates [13, 36, 64]; (2) SSGD based algorithm with second-order statistics, including adaptive gradient algorithms [55, 54] and algorithms for exploring the information from the gradient covariance matrix [51]; and (3) SSGD-based algorithms on a smoothed landscape [35, 9], in which specifically designed loss landscape smoothing algorithms are used. All of these approaches require global synchronization and/or global statistics collection, which makes them vulnerable to stragglers.

Decentralized algorithms, such as Decentralized Parallel Stochastic Gradient Descent (DPSGD) [33], are surrogates for SSGD in machine learning. Unlike SSGD, where each learner updates its weights by taking a global average of all learners' weights, DPSGD updates each learner's weights by taking a partial average (i.e., across a subset of neighboring learners). In contrast to the existing variants of SSGD, DPSGD requires no additional calculation and no global synchronization. Traditionally DPSGD is a second-choice to SSGD, and is used only when the underlying computational resources are less homogeneous (i.e., a high latency network or computational devices running at different speeds). Little thought has been given to the question of whether there are any convergence benefits for DPSGD, especially in the large batch setting.

In this paper, we find that DPSGD [33] greatly improves large batch training performance, as illustrated by the green curves in Figure 1. Since DPSGD only uses a partial average of neighboring learners' weights, each learner's weights differ from the weights of other learners. The differing weights between learners are an additional source of noise in DPSGD training. The key difference between SSGD, SSGD with Gaussian noise (denoted as "SSGD*" in this paper) and DPSGD is the source of noise during the update, and this noise directly affects performance in deep learning. This naturally motivates us to ask *Why does decentralized training outperform synchronous training in the large batch setting?* More specifically, we try to understand whether these performance differences are caused by differences in noise. We answer this question from both theoretical and empirical perspectives. Our contributions are:

- We analyze the dynamics of DDL algorithms, including both SSGD and DPSGD. We show, both theoretically and empirically, that the *intrinsic noise* in DPSGD automatically adjusts the effective learning rate when the batch size is large to help convergence. Note that the intrinsic noise comes completely for free in the DPSGD algorithm, and we show that it has

a loss-landscape smoothing effect. Guided by our theoretical results, we also investigate training tasks where careful learning rate warmup schemes are required (e.g., Transformer models) [56, 42, 52] and find that DPSGD can work with a much shorter learning rate warmup period thus simplifying hyper-parameter tuning.

- We conduct extensive empirical studies of 18 CV, ASR, and NLP tasks with state-of-the-art CNN, LSTM, and Transformer models. Our experimental results demonstrate that DPSGD consistently outperforms SSGD, across application domains and Neural Network (NN) architectures in the large batch setting, *without any hyper-parameter tuning*. To the best of our knowledge, DPSGD is the only generic algorithm that can improve SSGD large batch training and shorten learning rate warmup period for this many models/tasks. Furthermore, unlike other solutions, DPSGD does not require global synchronization.

The remainder of this paper is organized as follows. Section 2 details the problem formulation and learning dynamics analysis of SSGD, SSGD*, and DPSGD; Section 3 and Section 4 detail the empirical results; Section 5 discusses related work; and Section 6 concludes the paper.

## 2  Analysis of stochastic learning dynamics in SSGD and DPSGD

We first formulate the dynamics of an SGD based learning algorithm with multiple ($n > 1$) learners indexed by $j = 1, 2, 3, ...n$ following the same theoretical framework established for a single learner [3]. At time (iteration) $t$, each learner has its own weight vector $\vec{w}_j(t)$, and the average weight vector $\vec{w}_a(t)$ is defined as: $\vec{w}_a(t) \equiv n^{-1} \sum_{j=1}^{n} \vec{w}_j(t)$. Each learner $j$ updates its weight vector according to the cross-entropy loss function $L^{\mu_j(t)}(\vec{w})$ for minibatch $\mu_j(t)$ that is assigned to it at time $t$. The size of the local minibatch is $B$, and the overall batch size for all learners is $nB$. Two multi-learner algorithms, SSGD and DPSGD, are described below.

**(1) Synchronous Stochastic Gradient Descent (SSGD):** In the synchronous algorithm, each learner $j \in [1, n]$ starts from the average weight vector $\vec{w}_a$ and moves along the gradient of its local loss function $L^{\mu_j(t)}$ evaluated at the average weight $\vec{w}_a$:

$$\vec{w}_j(t + 1) = \vec{w}_a(t) - \alpha \nabla L^{\mu_j(t)}(\vec{w}_a(t)), \tag{1}$$

where $\alpha$ is the learning rate.

**(2) Decentralized Parallel SGD (DPSGD):** In the DPSGD algorithm [33], each learner $j$ computes the gradient at its own local weight $\vec{w}_j(t)$. The learning dynamics follows:

$$\vec{w}_j(t + 1) = \vec{w}_{s,j}(t) - \alpha \nabla L^{\mu_j(t)}(\vec{w}_j(t)). \tag{2}$$

where $\vec{w}_{s,j}(t)$ is the starting weight set to be the average weight of a subset of "neighboring" learners of learner-$j$, which corresponds to the non-zero entries in the mixing matrix [2] defined in [33] (note that $\vec{w}_{s,j} = \vec{w}_a$ if all learners are included as neighbors).

By averaging over all learners, the learning dynamics for the average weight $\vec{w}_a$ for both SSGD and DPSGD can be written formally the same way as:

$$\vec{w}_a(t + 1) = \vec{w}_a(t) - \alpha \vec{g}_a, \tag{3}$$

where $\vec{g}_a = n^{-1} \sum_{j=1}^{n} \vec{g}_j$ is the average gradient and $\vec{g}_j$ is the gradient from learner-$j$. The difference between SSGD and DPSGD is the weight at which $\vec{g}_j$ is computed: $\vec{g}_j \equiv \nabla L^{\mu_j(t)}(\vec{w}_a(t))$ is computed at $\vec{w}_a$ for SSGD; $\vec{g}_j \equiv \nabla L^{\mu_j(t)}(\vec{w}_j(t))$ is computed at $\vec{w}_j$ for DPSGD. The deviation of the weight for learner-$j$ from the average weight is defined as $\delta \vec{w}_j \equiv \vec{w}_j - \vec{w}_a$. It is easy to see that $\delta \vec{w}_j(t + 1) = \vec{w}_{s,j}(t) - \vec{w}_a(t) - \alpha [\vec{g}_j(t) - \vec{g}_a(t)]$, which depends on gradients at different points on the loss landscape.

### 2.1  Understanding DPSGD from the Optimization Perspective

The main difference between DPSGD and SSGD is that the stochastic gradients are calculated at different weights in DPSGD, while SSGD's stochastic gradient is calculated at the same weight. Intuitively, DPSGD explores more space than SSGD, which may help explain the empirical success of DPSGD. We formalize this intuition into the following theorem, which shows that DPSGD is optimizing a smoother landscape than SSGD.

---

[2]This is also called the "gossip matrix" in the literature, e.g., [27].

**Theorem 1.** *Denote $\mathcal{F}_t$ by the filtration generated by all the random variables until the $t$-th iteration. Suppose $n$ is large enough that $\left\| \frac{1}{n} \sum_{i=1}^{n} \nabla L^{\mu_i(t)}(\vec{w}_i(t)) - \frac{1}{n-1} \sum_{i=1}^{n-1} \nabla L^{\mu_i(t)}(\vec{w}_i(t)) \right\| \leq \epsilon$*

*almost surely, and assume $\delta\vec{w}_i(t) | \mathcal{F}_{t-1} \stackrel{i.i.d.}{\sim} \mathcal{N}(0, \sigma_w^2 I)$ with $i = 1, \ldots, n - 1$. Then from the $(t-1)$-th iteration to $t$-th iteration, SSGD and DPSGD are doing one step of stochastic gradient descent on two different functions $L(\vec{w})$ and $\tilde{L}(\vec{w}) \equiv \mathbb{E}_{\delta\vec{w}_i(t)}[L(\vec{w} + \delta\vec{w}_i(t)) \,|\, \mathcal{F}_{t-1}]$, respectively. The DPSGD loss $\tilde{L}(\vec{w})$ is smoother than the SSGD loss $L(\vec{w})$ if $L(\vec{w})$ is Lipschitz continuous.*

**Remark**: The proof of Theorem 1 can be found in Appendix A. Here, we briefly mention its implications. A function $f$ is defined as $l_s$-smooth if $\|\nabla f(x) - \nabla f(y)\| \leq l_s\|x - y\|$ for any $x, y$, where $l_s$ is the smoothness parameter of $f$. The landscape of the function $f$ is smoother when $l_s$ is smaller. Assume $L(\vec{w})$ is $G$-Lipschitz continuous, i.e., $|L(\vec{w}) - L(\vec{v})| \leq G\|\vec{w} - \vec{v}\|$, then by using Lemma 2 of [39], we know that the DPSGD landscape $\tilde{L}(\vec{w})$ is $\frac{2G}{\sigma_w}$-smooth. According to the convergence theory of SGD and DPSGD for nonconvex functions [11, 33, 12], the largest learning rate one can choose to guarantee convergence is $\frac{1}{l_s}$. For SSGD with the original loss landscape $L$, $l_s$ can be very large (even close to $+\infty$ due to the nonsmooth nature of the ReLU activation) while $l_s$ of the smoothed loss function $\tilde{L}$ for DPSGD is much smaller. This explains why we can use a larger learning rate in DPSGD as the landscape DPSGD sees has a smaller gradient-Lipschitz constant $l_s$ than that in SSGD.

It is important to note that $l_s$ of the smoothed loss function $\tilde{L}$ in DPSGD depends on the standard deviation $\sigma_w$ of weights from different learners. Since $\sigma_w$ depends on the loss landscape and changes with time (see Fig. 2(b)), the smoothing effect in DPSGD is self-adjusting – it is strong in the initial stage of training when the loss landscape is rough and becomes weaker as training progresses when the loss landscape becomes smoother. Our theoretical result suggests that this self-adjusting smoothing effect is responsible for DPSGD's convergence with a large learning rate in the large batch size setting. Next, we elaborate on this insight and verify it in a simple network for classification using the MNIST dataset.

Note that the Theorem 1 is only a one-step analysis. People may be interested in extending the analysis to trajectory-based analysis. We provide a sketch here. If we consider the perturbed objective $\tilde{L}(w) = \mathbb{E}_\delta[L(w + \delta)]$, where $\delta$ comes from the intrinsic noise of DPSGD, then we can utilize the descent lemma as shown in [11] to prove that DPSGD can converge to a stationary point of $\tilde{L}(w)$ in polynomial time. However, without the inherent noise of DPSGD, the landscape is rough and that is the reason why SSGD diverges. SSGD may not be able to converge to the stationary point of $L(w)$ (since the large learning rate in large batch setting makes the descent lemma not applicable in this case) or $\tilde{L}(w)$ (since there is no noise and landscape-smoothing effect in SSGD, so SSGD does not optimize the smoothed landscape). This is also consistent with our empirical evidence.

## 2.2 DPSGD Introduces a Landscape-Dependent Self-Adjusting Learning Rate that Helps Convergence

To understand the implication of the smoothing effect in DPSGD (Theorem 1) for learning dynamics, we define an effective learning rate $\alpha_e \equiv \alpha\vec{g}_a \cdot \vec{g}/\|\vec{g}\|^2$ by projecting the weight displacement vector $\Delta\vec{w}_a \equiv \alpha\vec{g}_a$ onto the direction of the gradient $\vec{g} \equiv \nabla L(\vec{w}_a)$ of the original loss function $L$ at $\vec{w}_a$. The learning dynamics, Eq. 3, can be rewritten as:

$$\vec{w}_a(t+1) = \vec{w}_a(t) - \alpha_e\vec{g} + \vec{\eta}_\perp, \tag{4}$$

where the "noise" term $\vec{\eta}_\perp \equiv -\alpha\vec{g}_a + \alpha_e\vec{g}$ describes the random weight dynamics in directions orthogonal to $\vec{g}$. The noise term has zero mean $\langle\vec{\eta}_\perp\rangle_\mu = 0$ and the noise strength is characterized by its variance $\Delta(t) \equiv \|\vec{\eta}_\perp\|^2$.

The effective learning rate $\alpha_e$ is related to the noise strength: $\alpha_e^2 = (\alpha^2\|\vec{g}_a\|^2 - \Delta)/\|\vec{g}\|^2$, which indicates that a higher noise strength $\Delta$ leads to a lower effective learning rate $\alpha_e$. The DPSGD noise $\Delta_{DP}$ is larger than the SSGD noise $\Delta_S$ by an additional noise term $\Delta^{(2)}(> 0)$ that originates from the difference of local weights $(\vec{w}_j)$ from their mean $(\vec{w}_a)$: $\Delta_{DP} = \Delta_S + \Delta^{(2)}$, see Appendix B for

details. By expanding $\Delta^{(2)}$ w.r.t. $\delta\vec{w}_j$, we obtain the average $\Delta^{(2)}$ over minibatch ensemble $\{\mu\}$:

$$
\begin{aligned}
\langle\Delta^{(2)}\rangle_\mu &\equiv \alpha^2 \langle||n^{-1}\sum_{j=1}^{n}[\nabla L^{\mu_j}(\vec{w}_j) - \nabla L^{\mu_j}(\vec{w}_a)]||^2\rangle_\mu \\
&\approx \alpha^2 \sum_{k,l,l'} H_{kl}H_{kl'}C_{ll'},
\end{aligned}
\tag{5}
$$

where $H_{kl} = \nabla^2_{kl}L$ is the Hessian matrix of the loss function and $C_{ll'} = n^{-2}\sum_{j=1}^{n}\delta w_{j,l}\delta w_{j,l'}$ is the weight covariance matrix. From Eq. 5 and the dependence of $\alpha_e$ on $\Delta$, it is clear that the effective learning rate in DPSGD depends directly on the loss landscape ($H$) and indirectly via the weight variance, $\sigma_w^2 = Tr(C)$, which decreases as the loss landscape becomes smooth (see Fig. 2(b)).

It is important to stress that the noise $\vec{\eta}_\perp$ in Eq.4 is not an artificially added noise. It is intrinsic to the use of minibatches (random subsampling) in all SGD-based algorithms (including SSGD and DPSGD). The noise is increased in DPSGD due to the weight difference among different learners ($\delta\vec{w}_j$). The noise strength $\Delta$ varies in weight space via its dependence on the loss landscape, as explicitly shown in Eq. 5. However, besides its landscape dependence, SGD noise scales inversely with the minibatch size $B$ [3]. With $n$ synchronized learners, the noise in SSGD scales as $1/(nB)$, which is too small to be effective for a large batch size $nB$. A main finding of our paper is that the additional landscape-dependent noise $\Delta^{(2)}$ in DPSGD can make up for the small SSGD noise when $nB$ is large and help enhance convergence in the large batch setting.

The landscape dependent smoothing effect in DPSGD (shown in Sec. 2.1) indicates that $\alpha_e$ in DPSGD is reduced at the beginning of training when the landscape is rough. To demonstrate effects of the landscape-dependent self-adjusting learning rates, we did detailed analysis in numerical experiments using the MNIST dataset. In this experiment, we used $n = 5$ learners with each learner a fully connected network with two hidden layers (50 units per layer) and we used $\vec{w}_{s,j} = \vec{w}_a$ for DPSGD. We focused on the large batch setting using $nB = 2000$ and a large learning rate $\alpha = 1$. As shown in Fig. 2(a), DPSGD converges to a solution with a low loss (2.1% test error), but SSGD fails to converge.

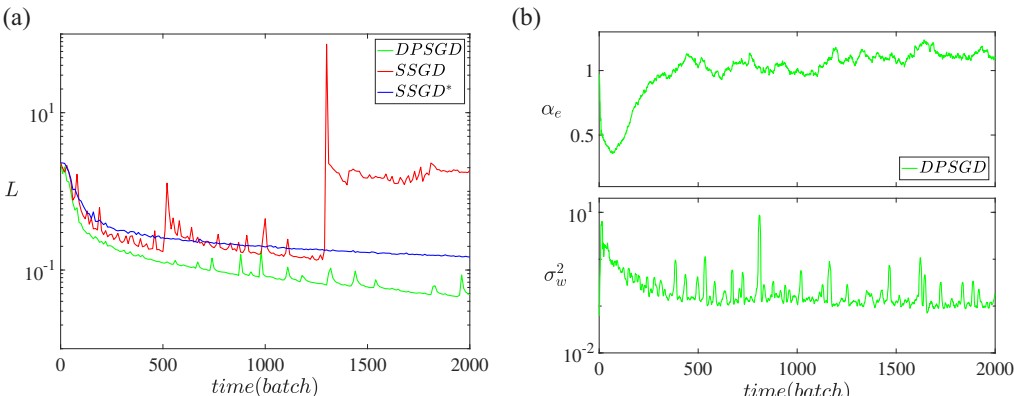

Figure 2: (a) Comparison of different multi-learner algorithms, DPSGD (green), SSGD (red), and SSGD* (blue) for a large learning rate $\alpha = 1$. The adaptive learning rate allows DPSGD to converge while SSGD fails to converge. A fine-tuned SSGD* also converges but to an inferior solution. (b) The effective learning rate for DPSGD $\alpha_e(DPSGD)$ is self-adaptive to the landscape – it is reduced in the beginning of training when gradients are large and recovers to $\sim \alpha$ when the gradients are small. The weight variance $\sigma_w^2(t)$ has the opposite landscape-dependence as $\alpha_e$ and decreases with training time.

To understand the convergence in DPSGD, we computed the effective learning rate ($\alpha_e$) and the weight variance ($\sigma_w^2$) during training. As shown in Fig. 2(b) (upper panel), the effective learning rate $\alpha_e$ is reduced in DPSGD during early training ($0 \le t \le 700$). This reduction of $\alpha_e$ is caused by the stronger noise $\Delta^{(2)}$ in DPSGD (see Fig. 4 in Appendix B), which is essential for convergence when gradients are large in the beginning of the training process. In the later stage of the training process when gradients are smaller, the landscape-dependent DPSGD noise decreases and $\alpha_e$ *automatically* increases back to be $\approx \alpha$ to allow fast convergence. From Eq. 5, the landscape-dependent noise in

|  |  | AlexNet | VGG | VGG-BN |
|---|---|---|---|---|
| bs=256 lr=1x | Baseline lr=0.01 | 56.31/79.05 | 69.02/88.66 | 70.65/89.92 lr=0.1 |
| bs=2048 lr=8x | SSGD DPSGD | **54.29/77.43** 53.71/76.91 | **67.67/87.91** 67.28/87.58 | **70.36/89.58** 69.76/89.31 |
| bs=4096 lr=16x | SSGD DPSGD | 0.10/0.50 **52.53/76.01** | 0.10/0.50 **66.44/87.20** | 65.39/86.51 **68.86/88.82** |
| bs=8192 lr=32x | SSGD DPSGD | 0.10/0.50 **49.01/73.00** | 0.10/0.50 **65.00/86.11** | 0.10/0.50 **63.55/85.43** |

Table 1: ImageNet-1K Top-1/Top-5 model accuracy (%) comparison for batch size 2048, 4096 and 8192. All experiments are conducted on 16 GPUs (learners), with batch size per GPU 128, 256 and 512 respectively. Bold text represents the best model accuracy achieved given the specific batch size and learning rate. The batch size 256 baseline is presented for reference. bs stands for batch-size, lr stands for learning rate. Baseline lr is set to 0.01 for AlexNet and VGG11, 0.1 for the other models. In the large batch setting, we use learning rate warmup and linear scaling as prescribed in [12]. For rough loss landscape like AlexNet and VGG, SSGD diverges when batch size is large whereas DPSGD converges.

DPSGD depends on the weight variance. As shown in Fig. 2(b) (lower panel), the weight variance $\sigma_w^2$ has a time-dependent trend that is opposite to $\alpha_e$: $\sigma_w^2$ is large in the beginning of training when the landscape is rough and decreases as training progresses and the landscape becomes smoother.

To show the importance of the landscape-dependent weight variance, we used SSGD*, which injects a Gaussian noise with a constant variance to weights in SSGD, i.e., by setting $\delta\vec{w}_j \overset{i.i.d.}{\sim} \mathcal{N}(0, \sigma_0^2 I)$ with a constant $\sigma_0^2$. We found that SSGD* fails to converge for most choices of noise strength $\sigma_0^2$. Only by fine tuning $\sigma_0^2$ can SSGD* converge, but to an inferior solution with much higher loss and test error (5.7%) as shown in Fig. 2(a).

Finally, in addition to helping convergence, we found that the landscape-dependent noise in DPSGD can also help find flat minima with better generalization in the large batch setting (see Appendix C for details).

## 3   Experimental Methodology

We implemented SSGD and DPSGD using PyTorch, OpenMPI, and NVidia NCCL. We ran experiments on a cluster of two 8-V100-GPU x86 servers. For CV tasks, we evaluated on CIFAR-10 (50,000 training samples, 178MB) and ImageNet-1K (1.2 million training samples, 140GB). For ASR tasks, we evaluated on SWB-300 (300 hours training data, 4,000,000 samples, 30GB) and SWB-2000 (2000 hours training data, 30,000,000 samples, 216GB). For the NLP task, we evaluated on Wikitext-103(103 million tokens, 180MB). In all, we evaluate 18 state-of-the-art NN models: 15 CNN models, 2 6-layer bi-directional LSTM models, and 1 16-layer GPT-2 transformer model. We summarize the model sizes and training times in Table 6 of Appendix D. Also refer to Appendix D for hardware configuration, software implementation, dataset and Neural Network (NN) model details.

## 4   Experimental Results

All the large batch experiments are conducted on 16 GPUs (learners). Batches are evenly distributed among learners, e.g., with sixteen learners, each learner uses a local batch size that is one sixteenth the overall batch size. A learner randomly picks a neighbor with which to exchange weights in each DPSGD iteration [59].

### 4.1   SSGD and DPSGD Comparison on CV Tasks (CIFAR-10 and ImageNet-1K)

On ImageNet-1K we test 6 CNN models – AlexNet, VGG11, VGG11-BN, ResNet-50, ResNext-50 and DenseNet-161. Among them, AlexNet and VGG have rougher loss landscapes and can only work with smaller learning rates, while VGG11-BN, ResNet-50, ResNext-50, and DenseNet-161 have smoother loss landscapes thanks to the use of BatchNorm or Residual Connections, and thus can work with larger learning rates. We use the same baseline training recipe prescribed in [4]:

|  | SWB-300 | | |
|---|---|---|---|
|  | bs2048 | bs4096 | bs8192 |
| SSGD | 1.58 | 10.37 | 10.37 |
| DPSGD | 1.59 | 1.60 | 1.66 |
|  | SWB-2000 | | |
|  | bs2048 | bs4096 | bs8192 |
| SSGD | 1.46 | 1.46 | 10.37 |
| DPSGD | 1.45 | 1.47 | 1.47 |

Table 2: Heldout loss comparison for SSGD and DPSGD, evaluated on SWB-300 and SWB-2000. There are 32000 classes in this task, a held-out loss 10.37 (i.e. $ln^{32000}$) indicates a complete divergence. bs stands for batch size.

Figure 3: SSGD diverges when the learning rate warmup period is 75 iterations while DPSGD converges with a warmup period as short as 25 iterations. (Wikitext103, GPT-2)

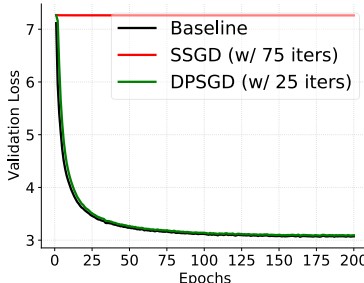

batch size 256, initial learning rate 0.01 for AlexNet and VGG-11 and 0.1 for the other 4 models, learning rate anneals by 0.1 every 30 epochs, 100 epochs in total. To study the model performance in the large batch setting, we follow the large batch size learning rate schedule prescribed in [12]: learning rate warmup for the first 5 epochs and then learning rate linear scaling w.r.t batch size. For example, in the AlexNet batch-size 8192 experiment, the learning rate is gradually warmed-up from 0.01 to 0.32 in the first 5 epochs, annealed to 0.032 from epoch 31 to epoch 60, annealed to 0.0032 from epoch 61 to epoch 90, and annealed to 0.00032 from epoch 91 to epoch 100. SSGD and DPSGD achieve comparable model accuracy in the large batch setting (see Table 10 in Appendix E.6). Most noticeably, when batch-size increases to 8192, SSGD diverges with AlexNet, VGG11, and VGG11-BN whereas DPSGD converges as shown in Table 1. Figure 9 in Appendix E.6 details the model accuracy progression versus epochs in each setting. Please see our detailed analysis of DPSGD vs SSGD on CIFAR-10 tasks throughout Appendix E.1 to Appendix E.5 where we document the DPSGD and SSGD comparison and loss landscape visualization (contour 2D projection and Hessian 2D projection), which show that DPSGD usually leads to much flatter optima than SSGD, and thus better generalization in the large batch setting.

*Summary* For rough loss landscapes like AlexNet and VGG, DPSGD converges whereas SSGD diverges in the large batch setting.

### 4.2 SSGD and DPSGD Comparison on ASR tasks

Unlike CV tasks where CNNs and their residual connection variants are the dominant models, ASR tasks overwhelmingly adopt RNN/LSTM models that capture sequence features. Furthermore, Batch-Norm is known not to work well in RNN/LSTM tasks [31]. Finally, there are over 32,000 different classes with wildy uneven distribution in our ASR tasks due to the Zipfian characteristics of natural language. All in all, ASR tasks present a much more challenging loss landscape than CV tasks to optimize over.

For the SWB-300 and SWB-2000 tasks, we follow the same learning rate schedule proposed in [57]: we use learning rate 0.1 for baseline batch size 256, and linearly warmup the learning rate w.r.t the baseline batch size for the first 10 epochs before annealing the learning rate by $\frac{1}{\sqrt{2}}$ for the remaining 10 epochs. For example, when using a batch size 2048, we linearly warmup the learning rate to 0.8 by the end of the 10th epoch before annealing. Table 2 illustrates heldout loss for SWB-300 and SWB-2000. In the SWB-300 task, SSGD diverges beyond batch size 2048 and DPSGD converges well until batch size 8192. In the SWB-2000 task, SSGD diverges beyond batch size 4096 and DPSGD converges well until batch size 8192. Figure 10 in Appendix E.7 details the heldout loss progression versus epochs.

*Summary* For ASR tasks, SSGD diverges whereas DPSGD converges to baseline model accuracy in the large batch setting.

### 4.3 Noise-injection and Learning Rate Tuning

In 6 out of 17 studied CV and ASR tasks, a large batch setting leads to a complete divergence in SSGD: EfficientNet-B0, AlexNet, VGG11, VGG11-BN, SWB-300 and SWB-2000. As discussed in

|  |  | AlexNet | VGG11 | VGG11-BN |
|---|---|---|---|---|
| lr*=32x | SSGD | 0.10/0.50 | 0.10/0.50 | 0.10/0.50 |
|  | DPSGD | 49.010/73.00 | **65.004/86.11** | 63.546/85.43 |
| lr=16x | SSGD | 0.10/0.50 | 0.10/0.50 | **70.11/89.47** |
|  | DPSGD | **49.26/73.14** | 62.046/83.98 | 69.108/89.07 |
| lr=8x | SSGD | 46.40/70.25 | 45.32/70.61 | 69.54/89.22 |
|  | DPSGD | 47.78/71.89 | 56.52/79.92 | 68.98/88.78 |
| lr=4x | SSGD | 41.77/66.44 | 50.20/74.83 | 68.61/88.57 |
|  | DPSGD | 42.18/66.96 | 48.52/73.33 | 67.98/88.22 |

Table 3: ImageNet-1K learning rate tuning for AlexNet VGG11, VGG11-BN with batch-size 8192. Bold text in each column indicates the best top-1/top-5 accuracy achieved across different learning rate and optimization method configurations for the corresponding batch size. DPSGD consistently delivers the most accurate models. *The learning rate 1x used here corresponds to batch size 256 baseline learning rate, and we still adopt the same learning rate warmup, scaling and annealing schedule. Thus 32x refers to linear learning rate scaling when batch size is 8192. By reducing learning rate to 16x, 8x and 4x, SSGD can escape early traps but still lags behind compared to DPSGD in most cases.

|  |  | SWB-300 (bs4096) | SWB-300 (bs8192) | SWB-2000 (bs 8192) |
|---|---|---|---|---|
| lr*=1.6/3.2 | SSGD | 10.37 | 10.37 | 10.37 |
|  | DPSGD | **1.60** | **1.66** | **1.47** |
| lr=0.8/1.6 | SSGD | 10.37 | 10.37 | 10.37 |
|  | DPSGD | 1.65 | 1.73 | 1.48 |
| lr=0.4/0.8 | SSGD | 1.76 | 10.37 | 1.51 |
|  | DPSGD | 1.77 | 1.80 | 1.52 |
| lr=0.2/0.4 | SSGD | 1.92 | 2.05 | 1.58 |
|  | DPSGD | 1.94 | 2.00 | 1.59 |

Table 4: Decreasing learning rate for SWB-300 and SWB-2000 (bs stands for batch-size). Bold text in each column indicates the best held-out loss achieved across different learning rate and optimization method configurations for the corresponding batch size. DPSGD consistently delivers the most accurate models. *learning rate 1.6 is used for bs4096 and learning rate 3.2 is used for bs8192. We still adopt the same learning rate warmup, scaling and annealing schedule (baseline learning rate is 0.1 for batch size 256).

Section 2, the intrinsic landscape-dependent noise in DPSGD effectively helps escape early traps (e.g., saddle points) and improves training by automatically adjusting the learning rate. In this section, we demonstrate these facts by systematically adding Gaussian noise (the same as the $SSGD^*$ algorithm in Section 2) and decreasing the learning rate. We find that SSGD might escape early traps but still results in a much inferior model compared to DPSGD.

**Noise-injection** In Figure 1, we systematically explore Gaussian noise injection with mean 0 and standard deviation (std) ranging from 10 to 0.00001 via binary search (i.e. roughly 20 configurations for each task). We found in the vast majority of the setups, noise-injection cannot escape early traps. In EfficientNet-B0, only when std is set to 0.04, does the model start to converge, but to a very low accuracy (test accuracy 22.15% in SSGD vs 91.13% in DPSGD). In the SWB-300 case, when std is 0.01, SSGD shows an early sign of converging for the first 3 epochs before it starts to diverge. In the AlexNet, VGG11, VGG11-BN, and SWB-2000 cases, we didn't find any configuration that can escape early traps. Figure 1 characterizes our best-effort Gaussian noise tuning and its comparison against SSGD and DPSGD. A plausible explanation is that Gaussian noise injection escapes saddle points very slowly, since Gaussian noise is isotropic and the complexity for finding local minima is dimension-dependent [10]. Deep Neural Networks are usually over-parameterized (i.e., high-dimensional), so it may take a long time to escape local traps. In contrast, the heightened landscape-dependent noise in DPSGD is anisotropic [3, 8] and can drive the system to escape in the right directions.

**Learning Rate Tuning** To make otherwise-divergent SSGD training converge in the large batch setting, we systematically tune down the learning rates. Table 3 and Table 4 compare the model quality trained by SSGD and DPSGD using smaller learning rates in the large batch setting, for ImageNet and

ASR tasks. Table 9 in Appendix E.3 illustrates the similar learning rate tuning effort for CIFAR-10 tasks. As we can see, by using a smaller learning rate, SSGD can escape early traps and converge, however it consistently lags behind DPSGD in the large batch setting. Morever, DPSGD does not depend on such an exhaustive learning rate tuning to achieve convergence. DPSGD can simply follow the learning rate warm-up and linear scaling rules [12] whereas SSGD requires much more stringent learning rate tuning. This implies DPSGD practitioners enjoy a much larger degree of freedom when it comes to hyper-parameter tuning in the large batch setting than the SSGD practitioners.

*Summary* By systematically introducing landscape-independent noise and reducing the learning rate, SSGD could escape early traps (e.g., saddle points), but results in much inferior models compared to DPSGD in the large batch setting.

## 4.4 DPSGD and SSGD Runtime Comparison

In Appendix F, we detail runtime comparison between DPSGD and SSGD and demonstrate DPSGD consistently runs faster than SSGD. We also compare DPSGD with LAMB[55], a state-of-the-art optimizer specifically designed for synchronous large-batch training, demonstrating that DPSGD can avoid straggler problems in distributed training.

## 4.5 SSGD and DPSGD Comparison on NLP tasks (Wikitext-103)

For NLP tasks such as Masked Language Modeling (MLM) [6, 50], a careful learning rate warmup scheme needs to be designed so that learning rate grows from 0 to a desired learning rate gradually. Too short a warmup period often leads to divergence and practitioners need to restart training, which wastes huge computational resources[42, 52, 56]. We test our theory by finding the shortest viable learning rate warmup period for SSGD and DPSGD. We use the hyper-parameter settings prescribed in [52], warmup learning rate 0 to $2.5 \times 10^{-4}$ in the first 64000 samples (i.e., 250 iterations of batch size 256) and then cosine-annealing to zero on top of an Adam optimizer. We then shorten the learning rate warmup period and check convergence. Figure 3 and Table 5 show that SSGD diverges when the learning rate warmup period is shorter than 100 iterations, while DPSGD converges with a warmup period as short as 25 iterations. Figure 1c shows that injecting independent random noise into SSGD (in the same fashion as Section 4.3) does not help SSGD escape early training traps. These experiments corroborate our theory that DPSGD can leverage loss landscape noise to self-adjust the learning rate.

| Warmup(iters) | 250 | 100 | 75 | 50 | 25 | 15 |
|---|---|---|---|---|---|---|
| SYNC | 3.09 | 3.07 | 7.26 | 7.26 | 7.26 | 7.26 |
| DPSGD | 3.08 | 3.053 | 3.06 | 3.08 | 3.09 | 7.26 |

Table 5: Validation loss comparison when shortening the learning rate warmup period. DPSGD can converge with a much shorter warmup. All experiments are conducted on 16 GPUs (learners). Wikitext-103, GPT-2 model, 200 epochs training in total.

## 5 Related Works

Please see Appendix G

## 6 Conclusion

In this paper, we find that in the large-batch and large-learning-rate setting, DPSGD yields comparable model accuracy when SSGD converges; moreover, DPSGD converges when SSGD diverges. We then investigate why DPSGD outperforms SSGD for large batch training. Through detailed analysis on small-scale tasks and an extensive empirical study of a diverse set of modern DL tasks, we conclude that the landscape-dependent noise, which is strengthened in the DPSGD system, self-adjusts the effective learning rate according to the loss landscape, helping convergence. This self-adjusting learning rate effect is a mere by-product of the inherent loss-landscape-dependent-noise of the DPSGD training algorithm and requires no additional computation, no additional communication and no additional hyper-parameter tuning. The theory was originally developed to understand why DPSGD outperforms SSGD in the large batch setting for CV and ASR tasks. The same theory can be also verified in NLP tasks where when a carefully designed learning rate warmup scheme is required.

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
