## A  Proof of Theorem 1

We first start to compare the learning dynamics of DPSGD and SSGD respectively. For DPSGD, we have

$$\vec{w}_a(t+1) = \vec{w}_a(t) - \alpha \cdot \frac{1}{n} \sum_{i=1}^{n} \nabla L^{\mu_i(t)}(\vec{w}_i(t)), \tag{6}$$

where $n$ is the number of machines, $i = 1, \ldots, n$ is the index of the machine, $\vec{w}_i(t)$ is the weight of the model at the $t$-th iteration on $i$-th machine, $\vec{w}_a(t) = \frac{1}{n} \sum_{i=1}^{n} \vec{w}_i(t)$, $L$ is the loss function, $\mu_i(t)$ denotes the minibatch sampled from the $i$-th machine at the $t$-th iteration, and $\alpha$ is the learning rate. In contrast, SSGD's update rule is

$$\vec{w}_a(t+1) = \vec{w}_a(t) - \alpha \cdot \frac{1}{n} \sum_{i=1}^{n} \nabla L^{\mu_i(t)}(\vec{w}_a(t)). \tag{7}$$

Define $\delta\vec{w}_i(t) = \vec{w}_a(t) - \vec{w}_i(t)$. Let us consider following fact: Given the realization of $\mu_i(t-1)$, $\vec{w}_i(t)$'s are mutually independent, and any $n-1$ random variables selected from $\{\delta\vec{w}_i(t)\}_{i=1}^{n}$ are mutually independent due to $\sum_{i=1}^{n} \delta\vec{w}_i(t) = 0$.

When $n$ is sufficiently large, we have the surrogate minibatch gradient with batch size $n - 1$ ($\frac{1}{n-1} \sum_{i=1}^{n-1} \nabla L^{\mu_i(t)}(\vec{w}_i(t))$) to be $\epsilon$-close to the minibatch gradient with size $n$ ($\frac{1}{n} \sum_{i=1}^{n} \nabla L^{\mu_i(t)}(\vec{w}_i(t))$), and hence can be regarded as approximate minibatch gradient with batch size $n - 1$, which are sampled i.i.d. from $\{\delta\vec{w}_i(t)\}_{i=1}^{n-1} \mid \mathcal{F}_{t-1}$. Once we have the independence, we can find that both (6) and (7) are doing SGD update, with different objective functions. In addition, assuming $\{\delta\vec{w}_i(t)\}_{i=1}^{n-1} \mid \mathcal{F}_{t-1}$ are i.i.d. Gaussian distribution is also reasonable due to the central limit theorem and the fact that $n$ is sufficiently large.

Then at the $t$-th iteration, (6) is using one step of SGD to optimize $L(\vec{w})$ directly, while (7) is using one step of SGD to optimize a smoothed version of $L$, which is $\mathbb{E}_{\delta\vec{w}_i(t)} [L(\vec{w} + \delta\vec{w}_i(t)) \mid \mathcal{F}_{t-1}]$.

Suppose $L(\vec{w})$ is $G$-Lipschitz continuous, by using Lemma 2 of [39], we know that the landscape DPSGD is trying to optimize over is $\tilde{L}(\vec{w})$ is $\frac{2G}{\sigma_w}$-smooth.

## B  Appendix for the Noise Analysis

To understand the origin of the noise term $\vec{\eta}$ in DPSGD, we decompose the gradient $\vec{g}_j$ for an individual learner-$j$:

$$\begin{aligned}
\vec{g}_j &= \vec{g}_0 + \delta g_j^{(1)} + \delta g_j^{(2)} \\
&= \nabla L^\mu(\vec{w}_a) + [\nabla L^{\mu_j}(\vec{w}_a) - \nabla L^\mu(\vec{w}_a)] \\
&\quad + [\nabla L^{\mu_j}(\vec{w}_j) - \nabla L^{\mu_j}(\vec{w}_a)],
\end{aligned} \tag{8}$$

where the first term $\vec{g}_0 \equiv \nabla L^\mu(\vec{w}_a)$ in the right hand side of Eq. 8 is the gradient of the loss function over the "superbatch" $\mu$ defined as the sum of all the minibatches for different learners at a given iteration: $\mu(t) = \sum_{j=1}^{n} \mu_j(t)$; the second term $\delta g_j^{(1)} \equiv \nabla L^{\mu_j}(\vec{w}_a) - \nabla L^\mu(\vec{w}_a)$ describes the gradient difference (fluctuation) between a minibatch $\mu_j$ and the superbatch $\mu$; the third term $\delta g_j^{(2)} \equiv \nabla L^{\mu_j}(\vec{w}_j) - \nabla L^{\mu_j}(\vec{w}_a)$ represents the difference (fluctuation) of the gradients at the individual weight $\vec{w}_j$ and at the average weight $\vec{w}_a$. Note that $\delta g_j^{(2)} = 0$ in SSGD as the gradients are taken at the average weight $\vec{w}_a$ for all learners. By taking the average of Eq. 8 over $j$, we have: $\vec{g}_a = \vec{g}_0 + \delta g_a^{(1)} + \delta g_a^{(2)}$ with $\delta g_a^{(i)} = n^{-1} \sum_{j=1}^{n} \delta g_j^{(i)}$ ($i = 1, 2$). Here, $\delta g_a^{(1)}$ vanishes after averaging over all minibatch. $\delta g_a^{(0)}$ is due to superbatch-superbatch difference and $\delta g_a^{(2)}$ comes from weight-weight difference in DPSGD. The gradient fluctuation has zero mean and its variance given by: $\Delta^{(2)} \equiv \alpha^2 \|\delta\vec{g}_a^{(2)}\|^2$. Finally, the noise strength in DPSGD $\Delta_{DP}$ can be expressed as:

$$\Delta_{DP} \equiv \|\vec{\eta}\|^2 = \Delta_S + \Delta^{(2)}, \tag{9}$$

where $\Delta_S \equiv \alpha^2 (\|\vec{g}_0\|^2 - (\vec{g}_0 \cdot \vec{g})^2 / \|\vec{g}\|^2)$ is the SSGD noise strength which is equivalent to the noise strength in a single-learner SGD algorithm with a superbatch (size $nB$). The $\Delta^{(2)}$ term only

577 exists in DPSGD. In general, this additional contribution makes the learning noise larger in DPSGD
578 than that in SSGD, although noise strength also depends on $\vec{g_a}$, $\vec{g_0}$, etc., which may be different for
579 different algorithms.

580 In Fig. 4, we calculated these two noise components of DPSGD for the experiment shown in Fig. 2.
581 Due to the large batch size we used in the experiment, $\Delta_S$ is very small during the training process.
582 However, the additional landscape-dependent noise $\Delta^{(2)}$ in DPSGD can make up for the small SSGD
583 noise when $nB$ is large and adaptively adjust the effectively learning rate $\alpha_e$ according to the loss
584 landscape to help convergence. This additional landscape dependent noise in SGD is also responsible
585 for finding flat minima with good generalization performance as shown in Fig. 5 in Appendix C.

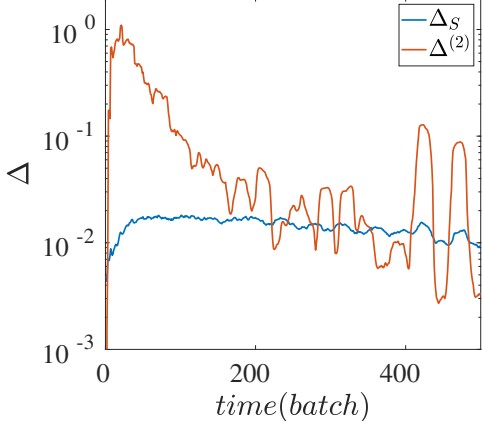

Figure 4: The noise in DPSGD can be decomposed into the SSGD noise $\Delta_S$ evaluated at the mean
weight $\vec{w}_a$ plus an additional noise $\Delta^{(2)}(> 0)$. The additional DPSGD noise $\Delta^{(2)} \gg \Delta_S$ in the
beginning of the training before it decreases to become comparable to $\Delta_S$.

## C  Appendix for the effect of DPSGD noise in help finding flat minima with better generalization

588 To demonstrate the effect of the additional noise in DPSGD for finding flat minima, we consider a
589 numerical experiment with a smaller learning rate $\alpha = 0.2$ for the MNIST dataset. We used $n = 6$
590 and $\vec{w}_{s,j}(t)$ in DPSGD is the average weight of 2 neighbors on each side. In this case, both SSGD
591 and DPSGD can converge to a solution, but their learning dynamics are different. As shown in Fig. 5
592 (upper panel), while the training loss $L$ of SSGD (red) decreases smoothly, the DPSGD training loss
593 (green) fluctuates widely during the time window (1000-3000) when it stays significantly above the
594 SSGD training loss. As shown in Fig. 5 (lower panel), these large fluctuations in $L$ are caused by the
595 high and increasing noise level in DPSGD. This elevated noise level in DPSGD allows the algorithm
596 to search in a wider region in weight space. At around time 3000(batch), the DPSGD loss decreases
597 suddenly and eventually converges to a solution with a similar training loss as SSGD. However,
598 despite their similar final training loss, the DPSGD loss landscape is flatter (contour lines further
599 apart) than SSGD landscape. Remarkably, the DPSGD solution has a lower test error (2.3%) than the
600 test error of the SSGD solution (2.6%). We have also tried the SSGD* algorithm, but the performance
601 (3.9% test error) is worse than both $SSGD$ and $DPSGD$.

602 To understand their different generalization performance, we studied the loss function landscape
603 around the SSGD and DPSGD solutions. The contour plots of the loss function $L$ around the two
604 solutions are shown in the two right panels in Fig. 5. We found that the loss landscape near the DPSGD
605 solution is flatter than the landscape near the SSGD solution despite having the same minimum
606 loss. Our observation is consistent with [24] where it was found that SSGD with a large batch size
607 converges to a sharp minimum which does not generalize well. Our results are in general agreement
608 with the current consensus that flatter minima have better generalization [16, 17, 1, 2, 63]. It was
609 recently suggested that the landscape-dependent noise in SGD-based algorithms can drive the system
610 towards flat minima [8]. However, in the large batch setting, the SSGD noise is too small to be

| | WikiText-103 | CIFAR10 | | | | |
|---|---|---|---|---|---|---|
| | GPT-2 | EfficientNet-B0 | VGG-19 | ResNet-18 | DenseNet-121 | MobileNet |
| Size | 201.58MB | 11.11 MB | 76.45 MB | 42.63 MB | 26.54 MB | 12.27 MB |
| Time | 320Hr | 2.92 Hr | 1.08 Hr | 1.37 Hr | 5.48 Hr | 1.02 Hr |
| | CIFAR10 | | | | SWB300 | SWB2000 |
| | MobileNetV2 | ShuffleNet | GoogleNet | ResNext-29 | LSTM | LSTM |
| Size | 8.76 MB | 4.82 MB | 23.53 MB | 34.82 MB | 164.62 MB | 164.62 MB |
| Time | 1.96 Hr | 2.46 Hr | 5.31 Hr | 4.55 Hr | 26.88 Hr | 203.21 Hr |
| | ImageNet-1K | | | | | |
| | AlexNet | VGG | VGG-BN | ResNet-50 | ResNext-50 | DenseNet-161 |
| Size | 233.08 MB | 506.83 MB | 506.85 MB | 97.49 MB | 95.48 MB | 109.41 MB |
| Time | 190.67 Hr | 168.67 Hr | 204.27 Hr | 238.8 Hr | 341.33 Hr | 664.53 Hr |

Table 6: Evaluated workload model size and training time. Training time is measured when running on 1 V100 GPU. CIFAR-10 is trained with batch size 128 for 320 epochs. ImageNet-1K is trained with batch size 256 for 100 epochs. SWB-300 and SWB-2000 are trained with batch size 128 for 16 epochs.

effective. The additional landscape-dependent noise $\Delta^{(2)}$ in DPSGD, which also depends inversely on the flatness of the loss function (see Eq. 5), is thus critical for the system to find flatter minima in the large batch setting.

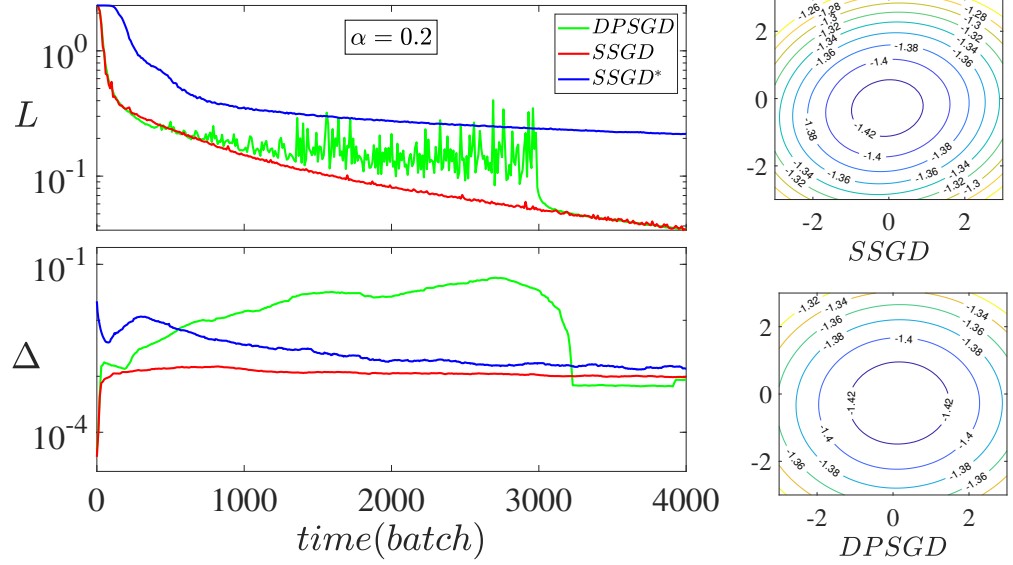

Figure 5: Comparison of different multi-learner algorithms, DPSGD (green), SSGD (red), and SSGD* (blue). For a smaller learning rate $\alpha = 0.2$, both SSGD and DPSGD converge, however, DPSGD finds a flatter minimum with a lower test error than SSGD. The fixed noise SSGD* has the worst performance. See text for detailed description.

# D  Appendix for Experimental Methodology

## D.1  Software and Hardware

We use PyTorch 1.6.0 (Torchvision 0.7.0) as the single learner DL engine. Our communication library is built with CUDA 11.0 compiler, the CUDA-aware OpenMPI 3.1.6, and g++ 8.5.0 compiler. Concurrency control of computation threads and communication threads is implemented via Pthreads. We run our experiments on a cluster of 8-V100 GPU servers. Each server has 2 sockets and 9 cores per socket. Each core is an Intel Xeon E5-2697 2.3GHz processor. Each server is equipped with 1TB main memory and 8 V100 GPUs. Between servers are 100Gbit/s Ethernet connections. GPUs and

CPUs are connected via PCIe Gen3 bus, which has a 16GB/s peak bandwidth in each direction per socket.

## D.2  Dataset and Models

We evaluate on three types of DL tasks: CV, ASR and NLP. For CV task, we evaluate on CIFAR-10 dataset [28], which comprises of a total of 60,000 RGB images of size $32 \times 32$ pixels partitioned into the training set (50,000 images) and the test set (10,000 images) and ImageNet-1K dataset [5], which comprises of 1.2 million training images (256x256 pixels) and 50,000 (256x256 pixels) testing images. We test CIFAR-10 with 10 representative CNN models [37]. The 10 CNN models are: (1) EfficientNet-B0, with a compound coefficient 0 in the basic EfficientNet architecture [49]. (3) VGG-19, a 19 layer instantiation of VGG architecture [46]. (4) ResNet-18, a 18 layer instantiation of ResNet architecture [14]. (5) DenseNet-121, a 121 layer instantiation of DenseNet architecture [20]. (6) MobileNet, a 28 layer instantiation of MobileNet architecture [19]. (7) MobileNetV2, a 19 layer instantiation of [45] architecture that improves over MobileNet by introducing linear bottlenecks and inverted residual block. (8) ShuffleNet, a 50 layer instantiation of ShuffleNet architecture [62]. (9) GoogleNet, a 22 layer instantiation of Inception architecture [48]. (10) ResNext-29, a 29 layer instantiation of [53] with bottlenecks width 64 and 2 sets of aggregated transformations. The detailed model implementation refers to [37]. Among these models, ShuffleNet, MobileNet, MobileNet-V2, EfficientNet represent the low memory footprint models that are widely used on mobile devices, where federated learnings is often used. The other models are standard CNN models that aim for high accuracy. We test 6 CNN models for ImageNet-1K, AlexNet [29], VGG11 [46], VGG11 with BatchNorm [21] VGG11-BN, ResNet-50 [14], ResNext-50 [53], and DenseNet-161 [20].

For ASR tasks, we evaluate on SWB-300 and SWB-2000 dataset. The input feature (i.e. training sample) is a fusion of FMLLR (40-dim), i-Vector (100-dim), and logmel with its delta and double delta (40-dim $\times 3$). SWB-300, whose size is 30GB, contains roughly 300 hour training data of over 4 million samples. SWB-2000, whose size is 216GB, contains roughly 2000 hour training data of over 30 million samples. The size of SWB-300 held-out data is 0.6GB and the size of SWB-2000 held-out data is 1.2GB. The acoustic model is a long short-term memory (LSTM) model with 6 bi-directional layers. Each layer contains 1,024 cells (512 cells in each direction). On top of the LSTM layers, there is a linear projection layer with 256 hidden units, followed by a softmax output layer with 32,000 (i.e. 32,000 classes) units corresponding to context-dependent HMM states. The LSTM is unrolled with 21 frames and trained with non-overlapping feature subsequences of that length. This model contains over 43 million parameters and is about 165MB large.

For NLP task, we evaluate on wikitext-103 dataset [38]. The model architecture is GPT-2 [43], with 16 attention layers, 256 sequence length, 10 attention heads, 410-dimension word embedding , and 2100 hidden dimensions. The vocab size is 28996. Model size is 201.58 MB.

Table 6 summarizes the model size and training time (on 1 V100 GPU) for evaluated tasks. CIFAR-10 tasks train 320 epochs, ImageNet-1K tasks train 100 epochs, and all ASR tasks train 16 epochs.

# E  Appendix for Results Section

## E.1  CIFAR-10 Single Learner Baseline

For CIFAR-10 experiments, we use the hyper-parameter setup proposed in [37]: a baseline 128 sample batch size and learning rate 0.1 for the first 160 epochs, learning rate 0.01 for the next 80 epochs, and learning rate 0.001 for the remaining 80 epochs. Using the same learning rate schedule, we keep increasing the batch size up to 8192. Table 7 in Appendix E records test accuracy under different batch sizes. Model accuracy consistently deteriorates beyond batch size 1024 because the learning rate is too small for the decreased number of parameter updates.

## E.2  SSGD and DPSGD Comparison on CIFAR-10

To improve model accuracy beyond batch size 1024, we apply the linear scaling rule (i.e., linearly increase learning rate w.r.t batch size) [14, 12, 60]. We use learning rate 0.1 for batch size 1024, 0.2 for batch size 2048, 0.4 for batch size 4096, and 0.8 for batch size 8192 (except in EfficientNet-B0 batchsize 8192, we use learning rate 0.7). Table 8 compares SSGD and DPSGD performance running

|  | Batch Size | | | | | | |
| --- | --- | --- | --- | --- | --- | --- | --- |
|  | 128 | 256 | 512 | 1024 | 2048 | 4096 | 8192 |
| EfficientNet-B0 | 87.51 | 89.32 | 91.28 | **91.92** | 90.62 | 88.00 | 84.85 |
| VGG-19 | 93.51 | **93.78** | 93.35 | 93.12 | 92.64 | 91.82 | 87.76 |
| ResNet-18 | **95.44** | 95.26 | 95.08 | 94.59 | 94.96 | 92.98 | 91.24 |
| DenseNet-121 | 95.06 | 95.27 | **95.42** | 95.11 | 94.81 | 93.09 | 92.34 |
| MobileNet | 89.53 | 90.96 | **92.39** | 92.24 | 91.22 | 89.54 | 86.59 |
| MobileNetV2 | 90.52 | 92.93 | 94.17 | **94.99** | 93.71 | 91.97 | 89.81 |
| ShuffleNet | 90.4 | 92.27 | 92.82 | **93.15** | 91.94 | 90.59 | 87.81 |
| GoogleNet | 94.99 | 95.06 | 94.97 | **95.32** | 94.05 | 92.78 | 91.09 |
| ResNext-29 | 95.35 | **95.66** | 95.31 | 95.42 | 94.24 | 93.00 | 91.06 |

Table 7: CIFAR-10 accuracy (%) with different batch size. Across runs, learning rate is set as 0.1 for first 160 epochs, 0.01 for the next 80 epochs and 0.001 for the last 80 epochs. Model accuracy consistently deteriorates when batch size is over 1024. Bold text in each row represents the highest accuracy achieved for the corresponding model, e.g., EfficientNet-B0 achieves highest accuracy at 91.92% with batch size 1024.

|  |  | Eff-B0 | VGG | Res-18 | Dense-121 | Mobile | MobileV2 | Shuffle | Google | ResNext-29 |
| --- | --- | --- | --- | --- | --- | --- | --- | --- | --- | --- |
| bs=128 lr=0.1 | Baseline | 87.51 | 93.51 | 95.44 | 95.06 | 89.53 | 90.52 | 90.40 | 94.99 | 95.35 |
| bs=1024 lr=0.1 | SSGD | **91.92** | 93.12 | 94.59 | 95.11 | 92.24 | **94.99** | 93.15 | **95.32** | 95.42 |
|  | DPSGD | 91.69 | **93.15** | 94.98 | **95.12** | 92.52 | 94.36 | **93.55** | 95.18 | **95.72** |
| bs=2048 lr=0.2 | SSGD | **91.69** | 92.64 | **94.96** | 95.11 | 91.72 | 94.24 | **92.91** | 94.76 | 94.19 |
|  | DPSGD | 91.06 | **93.05** | 94.86 | **95.32** | 92.72 | **94.51** | 92.89 | **94.80** | **95.30** |
| bs=4096 lr=0.4 | SSGD | **91.62** | 92.68 | 94.30 | 94.72 | 91.68 | **94.25** | **92.67** | 94.36 | 93.21 |
|  | DPSGD | 91.23 | **92.72** | **94.78** | **95.24** | **92.03** | 94.12 | 92.20 | **94.99** | **94.32** |
| bs=8192 lr=0.8 | SSGD | 10 | 87.11 | 92.70 | 92.79 | 91.10 | **93.22** | 92.09 | 93.72 | 92.38 |
|  | DPSGD | **91.13** | 90.52 | **94.34** | **94.79** | **91.80** | 93.09 | **92.36** | **93.84** | **92.55** |

Table 8: DPSGD and SSGD comparison for CIFAR-10, batch size 2048, 4096 and 8192, with learning rate set as 0.2, 0.4 and 0.8 respectively. All experiments are conducted on 16 GPUs (learners), with batch size per GPU 128, 256 and 512 respectively. Bold texts represent the best model accuracy achieved given the specific batch size and learning rate. When batch size is 8192, DPSGD significantly outperforms SSGD. The batch size 128 baseline is presented for reference. bs stands for batch-size, lr stands for learning rate.

with 16 GPUs (learners). SSGD and DPSGD perform comparably up to batch size 4096. When the batch size increases to 8192, DPSGD outperforms SSGD in all but one case. Most noticeably, SSGD diverges in EfficientNet-B0 when the batch-size is 8192. Figure 6 in Appendix E.4 details the model accuracy progression versus epochs in each setting. To better understand the loss landscape in SSGD and DPSGD training, we visualize the landscape with 2D contour projections and 2D Hessian projections in Appendix E.5, using the method from [32]. Results in Appendix E.5 demonstrate that DPSGD can often find flatter optima than SSGD for CIFAR-10 tasks, which is consistent with results for MNIST shown in Appendix C. *Summary* DPSGD outperforms SSGD for 8 out of 9 CIFAR-10 tasks in the large batch setting. Moreover, SSGD diverges on the EfficientNet-B0 task. DPSGD is more effective at avoiding early traps and reaching better solutions than SSGD in the large batch setting.

### E.3   CIFAR-10 Hyper-Parameter Tuning

By reducing learning rate in the CIFAR-10 batchsize 8192 case as shown in Table 9, SSGD can escape early traps but still lags behind DPSGD. Bold text in each column indicates the best accuracy achieved for that model across different learning rate and optimization method configurations. DPSGD consistently delivers the most accurate models.

### E.4   CIFAR-10 Training Progression

Figure 6 illustrates SSGD and DPSGD comparison for CIFAR-10. SSGD and DPSGD perform comparably up to batch size 4096. When batch size increases up to 8192, DPSGD outperforms SSGD in all but one cases. Noticeably, SSGD diverges in EfficientNet-B0 when batch-size is 8192.

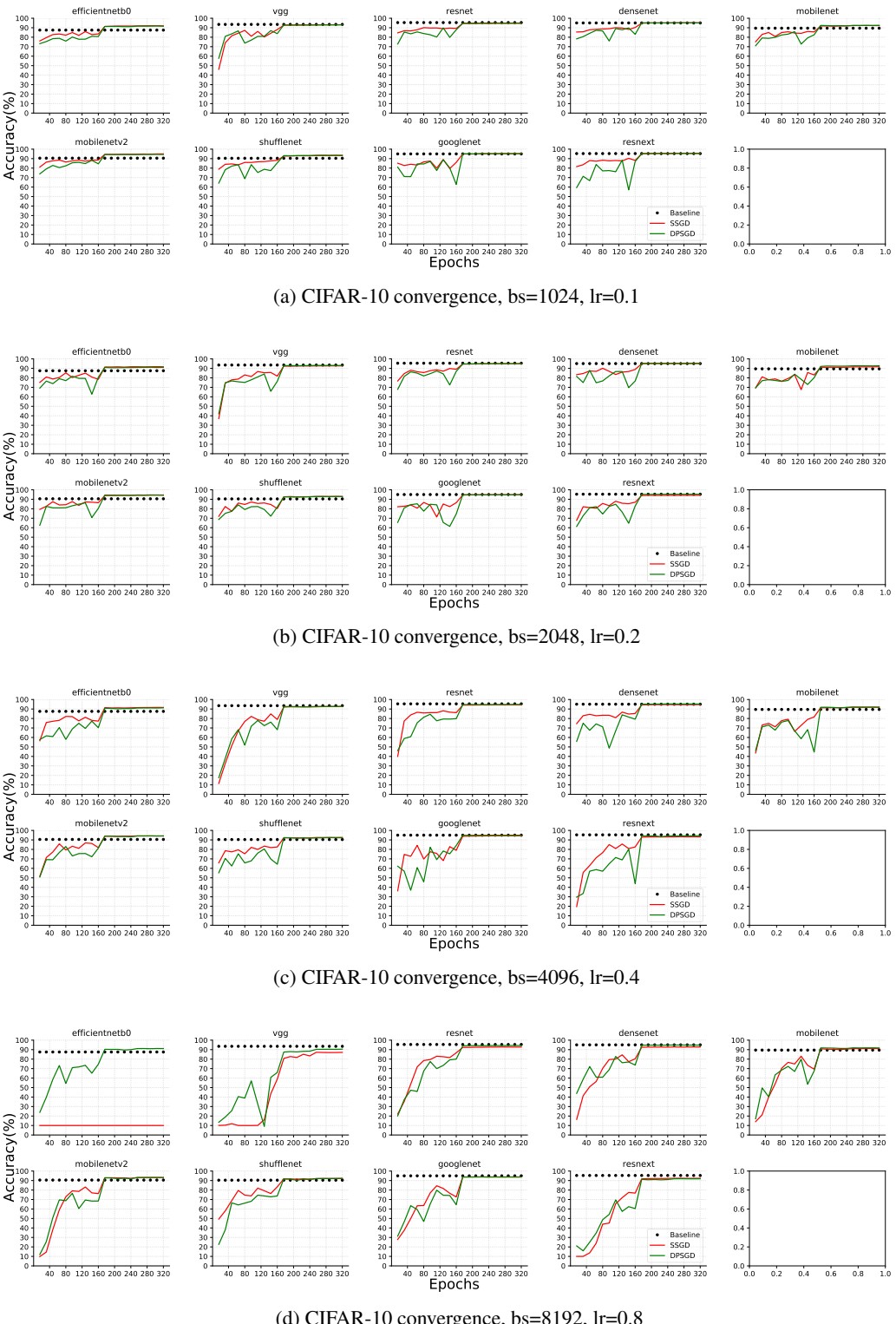

(a) CIFAR-10 convergence, bs=1024, lr=0.1

(b) CIFAR-10 convergence, bs=2048, lr=0.2

(c) CIFAR-10 convergence, bs=4096, lr=0.4

(d) CIFAR-10 convergence, bs=8192, lr=0.8

Figure 6: CIFAR-10 SSGD DPSGD comparison for batch size 2048, 4096 and 8192, with learning rate set as 0.2, 0.4 and 0.8 respectively. All experiments are conducted on 16 GPUs (learners), with batch size per GPU 128,256 and 512 respectively. When batch size is 8192, DPSGD significantly outperforms SSGD. bs stands for batch-size, lr stands for learning rate. The dotted black line represents the bs=128 baseline.

|         |       | Eff-B0 | VGG   | Res-18 | Dense-121 | Mobile | MobileV2 | Shuffle | Google | ResNext-29 |
|---------|-------|--------|-------|--------|-----------|--------|----------|---------|--------|------------|
| lr=0.8  | SSGD  | 10.00  | 87.11 | 92.7   | 92.79     | 91.10  | **93.22**| 92.09   | 93.72  | 92.38      |
|         | DPSGD | **91.13** | 90.52 | **94.34** | **94.79** | **91.80** | 93.09 | **92.36** | **93.84** | 92.55   |
| lr=0.4  | SSGD  | 88.61  | 91.06 | 91.98  | 93.42     | 91.13  | 93.11    | 91.54   | 92.85  | 89.70      |
|         | DPSGD | 89.80  | **91.93** | 93.91 | 94.32  | 91.38  | 93.14    | 91.68   | 93.49  | **92.79**  |
| lr=0.2  | SSGD  | 88.03  | 90.51 | 92.13  | 92.98     | 88.38  | 91.68    | 90.14   | 92.44  | 91.31      |
|         | DPSGD | 87.69  | 91.59 | 93.30  | 94.28     | 89.18  | 92.52    | 90.13   | 93.41  | 91.79      |

Table 9: CIFAR-10 with batch size 8192. By reducing learning rate, SSGD can escape early traps but still lags behind DPSGD. Bold text in each column indicates the best accuracy achieved for that model across different learning rate and optimization method configurations. DPSGD consistently delivers the most accurate models.

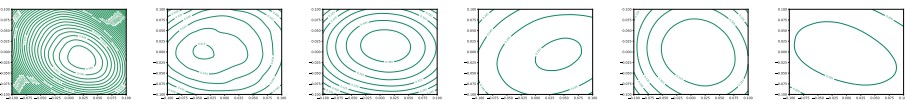

(a) VGG-S   (b) VGG-DP   (c) ResN-S   (d) ResN-DP   (e) DenseN-S   (f) DenseN-DP

Figure 7: CIFAR-10 2D contour plot. The more widely spaced contours represent a flatter loss landscape and a more generalizable solution. The distance between each contour line is 0.005 across all the plots. We plot against the model trained at the end of 320th epoch. VGG: VGG-19, ResN: ResNet-18, DenseN: DenseNet-121, -S: -SSGD, -DP: -DPSGD

### E.5 CIFAR-10 Loss Landscape Visualization

To better understand the loss landscape in SSGD and DPSGD training, we visualize the landscape contour 2D projection and Hessian 2D projection, using the same mechanism as in [32]. For both plots, we randomly select two $N$-dim vectors (where $N$ is the number of parameters in each model) and multiply with a scaling factor evenly sampled from -0.1 to 0.1 in a $K \times K$ grid to generate $K^2$ perturbations of the trained model. To produce a contour plot, we calculate the testing data loss of the perturbed model at each point in the $K \times K$ grid. Figure 7 depicts the 2D contour plot for representative models (at the end of the 320th epoch) in a $50 \times 50$ grid. DPSGD training leads not only to a lower loss but also much more widely spaced contours, indicating a flatter loss landscape and more generalizable solution. For the Hessian plot, we first calculate the maximum eigen value $\lambda_{\max}$ and minimum eigen value $\lambda_{\min}$ of the model's Hessian matrix at each sample point in a 4x4 grid. We then calculate the ratio $r$ between $|\lambda_{\min}|$ and $|\lambda_{\max}|$. The lower $r$ is, the more likely it is in a convex region and less likely in a saddle region. We then plot the heatmap of this $r$ value in this 4x4 grid. The corresponding models are trained at the 16-th epoch (i.e. the first 5% training phase) and the corresponding Hessian plot Figure 8 indicates DPSGD is much more effective at avoiding early traps (e.g., saddle points) than SSGD.

### E.6 ImageNet-1K Training Progression

Figure 9 illustrates SSGD and DPSGD comparison for ImageNet-1K. Noticeably, SSGD diverges in AlexNet, VGG11, VGG11-BN when batch-size is 8192 while DPSGD converges.

### E.7 SWB Training Progression

Figure 10 illustrates heldout loss comparison for SWB-300 and SWB-2000. In SWB-300 task, SSGD diverges beyond batch size 2048 and DPSGD converges well til batch size 8192. In SWB-2000 task, SSGD diverges beyond batch size 4096 and DPSGD converges well til at least batch size 8192.

## F   Appendix: End-to-End Run-time Comparison and Advice for Practitioners

**End-to-End Run-time Comparison**   In all above-mentioned DPSGD and SSGD experiments we used the *same* number of epochs as in the well-tuned single-GPU baseline (i.e., the total computation cost is fixed). When computation cost is fixed, DPSGD inherently runs faster than SSGD because DPSGD requires less messages transmitted and tolerate high-latency network better [33]. Table 11 records training time for each representative task (batch size 128 per GPU, 16 GPUs) on both low and

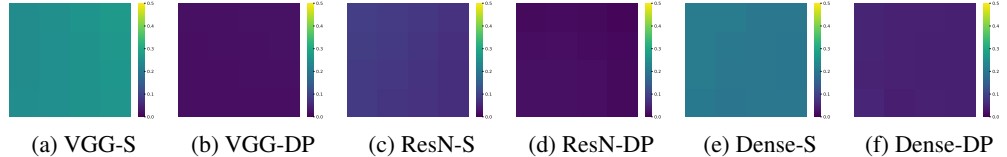

|  |  |  |  |  |  |
|---|---|---|---|---|---|
| (a) VGG-S | (b) VGG-DP | (c) ResN-S | (d) ResN-DP | (e) Dense-S | (f) Dense-DP |

Figure 8: CIFAR-10 Hessian heatmap on a 4x4 grid. The lower value (i.e. a cooler color) indicates the corresponding point is less likely in a saddle. We plotted against the models at the end of the 16th epoch. DPSGD is much more effective at avoiding early traps (e.g., saddle points) than SSGD. VGG: VGG-19, ResN: ResNet-18, DenseN: DenseNet-121, -S: -SSGD, -DP: -DPSGD

|  |  | AlexNet | VGG | VGG-BN | ResNet-50 | ResNext-50 | DenseNet-161 |
|---|---|---|---|---|---|---|---|
| bs=256 lr=1x | Baseline lr=0.01 | 56.31/79.05 lr=0.01 | 69.02/88.66 | 70.65/89.92 lr=0.1 | 76.39/93.05 | 77.62/93.64 | 78.43/94.20 |
| bs=2048 lr=8x | SSGD | **54.29/77.43** | **67.67/87.91** | **70.36/89.58** | **76.648/92.99** | **77.486/93.62** | **78.19/94.16** |
|  | DPSGD | 53.71/76.91 | 67.28/87.58 | 69.76/89.31 | 76.094/92.82 | 77.236/93.60 | 77.28/93.64 |
| bs=4096 lr=16x | SSGD | 0.10/0.50 | 0.10/0.50 | 65.39/86.51 | **76.46/93.06** | **77.43/93.65** | **77.98/93.86** |
|  | DPSGD | **52.53/76.01** | **66.44/87.20** | **68.86/88.82** | 75.784/92.82 | 77.24/93.54 | 77.73/93.81 |
| bs=8192 lr=32x | SSGD | 0.10/0.50 | 0.10/0.50 | 0.10/0.50 | **76.096/92.80** | 76.564/93.16 | **77.34/93.65** |
|  | DPSGD | **49.01/73.00** | **65.00/86.11** | **63.55/85.43** | 75.618/92.75 | **77.162/93.42** | 77.22/93.61 |

Table 10: ImageNet-1K Top-1/Top-5 model accuracy (%) comparison for batch size 2048, 4096 and 8192. All experiments are conducted on 16 GPUs (learners), with batch size per GPU 128, 256 and 512 respectively. Bold texts represent the best model accuracy achieved given the specific batch size and learning rate. The batch size 256 baseline is presented for reference. bs stands for batch-size, lr stands for learning rate. Baseline lr is set to 0.01 for AlexNet and VGG11, 0.1 for the other models. In the large batch setting, we use learning rate warmup and linear scaling as prescribed in [12]. For rough loss landscape like AlexNet and VGG, SSGD diverges when batch size is large whereas DPSGD converges.

high latency networks. Other tasks and batch-size setups show the same trend: DPSGD runs faster than SSGD. Further note that for Eff-B0 (target accuracy 90%) and SWB-2000 (target heldout loss 1.48), DPSGD reaches target model quality with twice the batch size as used in SSGD, all learning rates considered (Table 9 , Table 4). Thus DPSGD can effectively use 2X more GPUs. DPSGD achieves target accuracy for Eff-B0 in 0.067 hours and for SWB-2000 in 10.08 hours (64 GPUs). In contrast, SSGD achieves target accuracy for Eff-B0 in 0.19 hours and for SWB-2000 in 23.15 hours (32 GPUs).

In addition, DPSGD is immune to stragglers, while approaches that require global synchronization suffer slowdowns. Figure 11 demonstrates when there is a learner running 5x slower than other learners, DPSGD converges much faster than LAMB[55], a state-of-the art SSGD based large-batch training solution, on the SWB300 task. This experiment demonstrates that even SSGD-variant algorithms (e.g., LAMB) can be designed to work for specific training tasks, DPSGD can simultaneously tackle the convergence problem and straggler-avoidance problem for the generic large batch training tasks.

*Summary* DPSGD consistently runs faster than SSGD to reach target accuracy in the large batch setting.

|  |  | Eff-b0 | Res-18 | Dense-121 | Mobile | Google | ResNext-29 | SWB-2000 |
|---|---|---|---|---|---|---|---|---|
|  | Single-GPU | 2.92 | 1.37 | 5.48 | 1.02 | 5.31 | 4.55 | 203.21 |
| Latency (1$\mu s$) | SSGD | 0.34 | 0.35 | 0.68 | 0.17 | 0.58 | 0.56 | 38.00 |
|  | DPSGD | 0.26 | 0.32 | 0.58 | 0.12 | 0.49 | 0.41 | 29.71 |
| Latency (1$ms$) | SSGD | 0.46 | 0.82 | 0.96 | 0.30 | 0.84 | 0.94 | 96.31 |
|  | DPSGD | 0.27 | 0.32 | 0.58 | 0.13 | 0.50 | 0.42 | 29.85 |

Table 11: Time (hours) to complete training with batch size 128 per GPU and 16 GPUs in total (CIFAR-10 and SWB-2000).

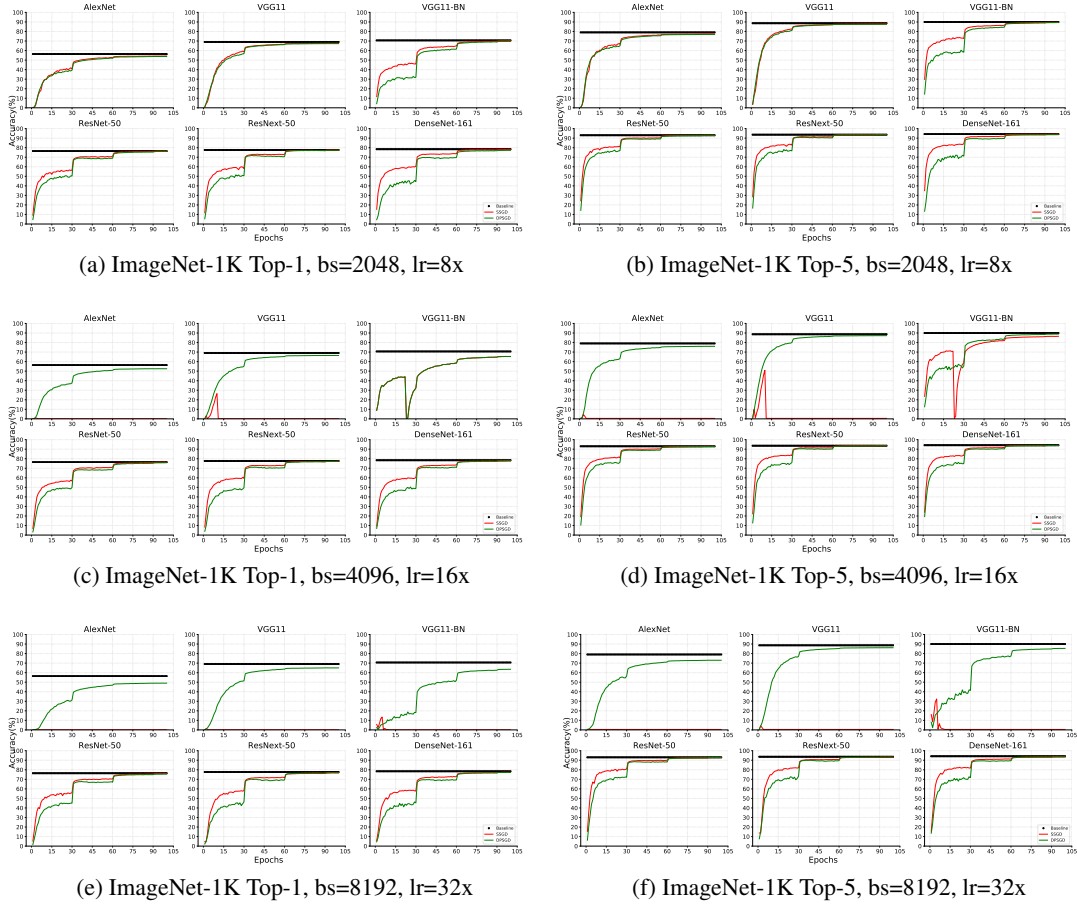

Figure 9: ImageNet-1K SSGD DPSGD comparison for batch size 2048, 4096 and 8192, with learning rate set as 0.2, 0.4 and 0.8 respectively. All experiments are conducted on 16 GPUs (learners), with batch size per GPU 128,256 and 512 respectively. When batch size is 8192, DPSGD significantly outperforms SSGD. bs stands for batch-size, lr stands for learning rate. The dotted black line represents the bs=256 baseline.

**Advice for Practitioners** In SSGD, when total batch size is fixed, the convergence behavior is the same regardless of the number of learners. In DPSGD, when the number of learners increases, the convergence could be harmed due to too much discrepancy between learners. In another word, we would like a system that has enough system noise so that it can help avoid early training traps but not too much noise so that model convergence is unaffected. In practice, we found that 16-learner setup usually yields the best convergence results in the DPSGD setting, which is consistent with research literature [33, 34]. To make use of a larger number of computing devices in DPSGD, we recommend a hierarchical system design [58] where we group nearby learners (e.g., on the same server) as one big super-learner and apply DPSGD algorithm only across super-learners. For example, on a 128 GPU cluster, we could group 8 learners as one big super-learner and we apply DPSGD among 16 super-learners. In addition, we also recommend in each iteration, each (super)-learner selects a random neighbor to communicate to further improve convergence. Please refer to [59] for the detailed analysis of how randomized communication improves DPSGD convergence.

## G  Related Work

To increase parallelism in DDL, one must increase batch size, which often leads to a deteriorating model accuracy [61, 30]. Meticulous task-specific learning rate tuning for large batch training exists in CV training [12, 54], NLP training [55] and ASR training [57]. Among them, layer-wise adaptive

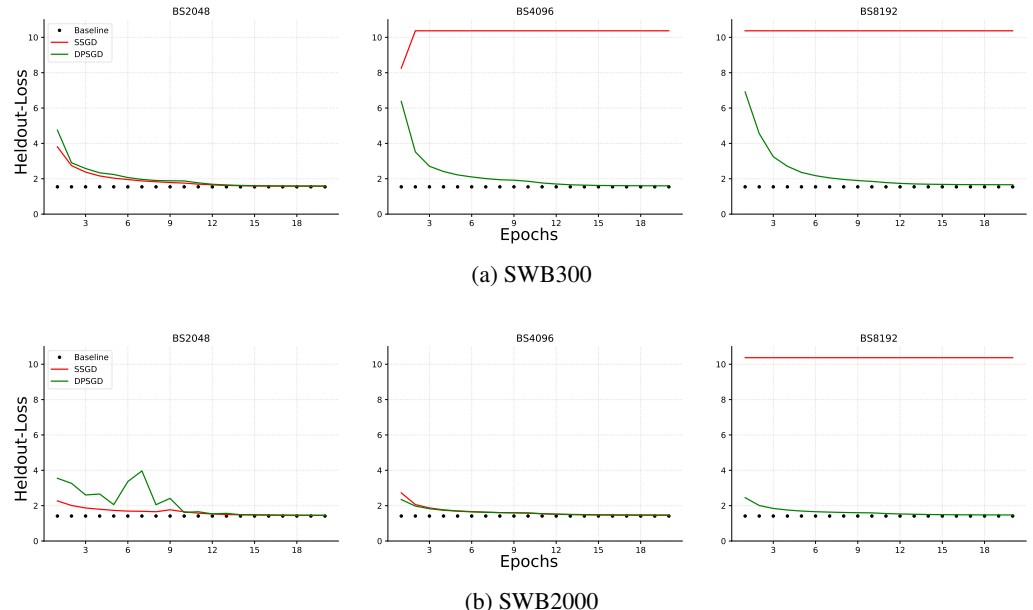

(a) SWB300

(b) SWB2000

Figure 10: Heldout loss w.r.t epochs for SWB-300 and SWB-2000. Dotted black lines indicate the batch size 256 heldout loss baseline.

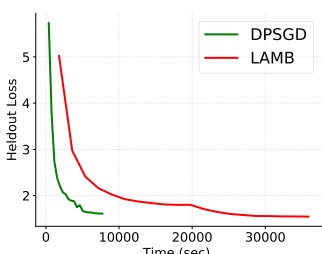

Figure 11: LAMB (a state-of-the-art SSGD based solution) and DPSGD comparison when there is a straggler that runs 5x slower than other learners in the system. SWB-300 task, batch size 4096, x-axis is running time and y-xais is the held-out loss.

learning rate tuning schemes [54, 55] rely on the Adam optimizer [25], which may diverge on some simple convex functions [44]. In particular, [54, 55] requires every learner to see other learner's gradients to calculate the large minibatch gradient, [9] optimizes both original loss function and the sharpness of the minimization, [35] calculates extra-gradient information and [51] leverages the covariance matrix of gradients noise. Furthermore, all above-mentioned approaches require global synchronization and suffer from the straggler problem: one slow learner can slow down the entire training process.The noise in the stochastic gradient plays an important role in terms of generalization performance in deep learning. Keskar et al. [24] show that large batch training procedures usually find sharp minima with poor generalization performance. This phenomenon is analyzed from different perspectives, including PAC-Bayesian learning theory [40, 41, 7], stochastic differential equation [22], Bayesian inference [47] and optimization theory [26]. There are several efforts trying to design algorithms to find flat minima that generalize better than SGD [2, 23].