# OpenReview forum: "Loss Landscape Dependent Self-Adjusting Learning Rates in Decentralized Stochastic Gradient Descent"
_NeurIPS.cc/2022/Conference — NeurIPS 2022 Submitted_

### Official Review · Reviewer_bt9N · 2022-07-09

**Rating:** 8
**Confidence:** 5
**Soundness:** 4 excellent
**Presentation:** 3 good
**Contribution:** 3 good

**Summary:**

The paper compares the convergence behavior of two common training schemes for distributed stochastic gradient descent optimization on large batch sizes. In particular, Synchronous Stochastic Gradient Descent (SSGD), where weight updates are computed by synchronizing all distributed learners globally, is being evaluated against Decentralized Parallel Stochastic Gradient Descent (DPSGD), where only a random selection of neighboring distributed learners is used to compute the weight update. The authors observe a much better convergence behavior with DPSGD and explore the hypothesis that this is due to an inherent quality of this training scheme to introduce stabilizing noise in the weight update rule, which automatically scales the learning rate depending on the landscape of the loss function. The paper continues to give a theoretical explanation for this observation and finally provides a very extensive evaluation on various machine learning tasks in a large distributed training environment. The results demonstrate a clear advantage of DPSGD for large batch sizes and a more robust convergence behavior that doesn't require to reduce the learning rates drastically, as it is the case for SSGD.

**Questions:**

It is not very clear to me why only Gaussian noise has been considered, it appears like the noise simulation could have been designed such that it is more similar to the noise introduced by DPSGD.

line 213: numbers up to twelve (12) should be written as words


**Limitations:**

The authors did not really discuss the limitations of their evaluation, which is unfortunate. As mentioned earlier, especially the simulated noise injection has a lot of room for further considerations. It again appears that this discussion has been omitted due to the page limit. An outlook on potential societal impact is also not provided. Since the important results of this paper mostly benefit groups with very large training resources, some discussion on this topic would have been appropriate.

**Strengths And Weaknesses:**

The claim of this paper is very adequately examined. The biggest strength of this paper is clearly the very deep evaluation, taking large CV, NLP and ASR tasks into account. The authors hereby build on top of previously published modern state-of-the-art training recipes with warm-up periods and adaptive gradient optimizers in order to ensure optimal performance. The thorough evaluation and the conclusions drawn from the experiments is convincing and of high significance to the field.

The scope of this work seems to be a bit too large for this conference. At various points in the paper it is noticeable that the authors were struggling with the page limit. It is particularly unfortunate that the description of related work had to be put into the appendix. One way to reduce the scope of this paper would be to leave out the evaluation of noise-injection, which appears rather limited by sticking to Gaussian noise injection at various points in the training process. While these results somehow support the claims by the authors, that the advantages of DPSGD are related to their inherent noise introduction, it is far from being proven to be the case given that manual noise injection experiments with SSGD contain many heuristics and best effort attempts to simulate the effect of DPSGD. In my opinion this could have been studied in a separate publication.

---

> ### Author Response · Authors · 2022-08-01
> **Response to Reviewer bt9N**
>
> Questions
>
> (1)  It is not very clear to me why only Gaussian noise has been considered, it appears like the noise simulation could have been designed such that it is more similar to the noise introduced by DPSGD.
>
> It is true that the noise introduced by DPSGD is not Gaussian and we don’t have an easy way to simulate DPSG-noise. The point of SSGD* experiment is to show that one cannot simply use Gaussian noise (even with fine-tuned noise strength) to get the effect of DPSGD noise. The good news is that one can get DPSGD noise for free in a DPSGD system.
>
> Limitations
>
> (1) The authors did not really discuss the limitations of their evaluation, which is unfortunate. As mentioned earlier, especially the simulated noise injection has a lot of room for further considerations. It again appears that this discussion has been omitted due to the page limit. An outlook on potential societal impact is also not provided. Since the important results of this paper mostly benefit groups with very large training resources, some discussion on this topic would have been appropriate.
>
> We fine tune SSGD* with different Gaussian noise strengths. In the future work, we can consider different type of noise distribution to see if it is possible to get some noise closer to DPSGD noise.
>
> One comment on societal impact: DPSGD can converge under a larger range of hyperparameters, so it can reduce the number  of very expensive training trials,  thus reducing energy cost.

---

### Official Review · Reviewer_2WDQ · 2022-07-09

**Rating:** 4
**Confidence:** 2
**Soundness:** 2 fair
**Presentation:** 2 fair
**Contribution:** 2 fair

**Summary:**

This work compares two stochastic gradient descent methods for Distributed Deep Learning (DDL), one is Synchronous Stochastic Gradient Descent (SSGD), and the other is Decentralized Parallel SGD (DPSGD), in a large batch setting. The authors empirically find that DPSGD not only has a runtime benefit but also a significant convergence benefit over SSGD in the large batch setting, which is due to additional landscape-dependent noise brought by DPSGD.

To understand this phenomenon, the authors perform both theoretical analysis and experimental investigation. They claim that the introduced noise smooths the loss landscape. They also conduct extensive studies over 18 state-of-the-art DL models/tasks and demonstrate that DPSGD often converges in cases where SSGD diverges when training is sensitive to large learning rates.

**Questions:**

(1) The orthogonal decomposition of noise in Equation (4) is based on the expectation of users. In other words, it only holds when the number of users is very large. However, the experiment shown in Figure 2 only performs with 5 learners. In this case, is the results of the experiment consistent with the theoretical analysis?


(2) In Table 1, the best performance comes from when lr=1x, and the performance drops significantly when lr is very large. What are the advantages of a large learning rate?

**Limitations:**

Even though the authors claim that they describe the limitations of their work. However, I cannot an explicit description. I would like to ask the authors if they can show me where is the limitation during the rebuttal.

**Strengths And Weaknesses:**

Strengths

(1) The authors provide a theoretical analysis to compare the smoothness of the two methods.

(2) This work conducts experiments on an adequate number of tasks and datasets.

Weaknesses

(1) The writing of this work has a big problem. The authors put important context including Related Works and DPSGD and SSGD Runtime Comparison in the appendix.

Related work is almost an indispensable chapter of an article. Besides, whether DPSGD has an advantage in runtime compared to SSGD is also important. Since the authors claim that DPSGD can have an advantage in terms of convergence, it is to be expected whether this advantage can bring about a saving in runtime. Although the conference has body content restrictions, putting very important content in the appendix is ​​a discouraged behavior.

(2) Although this article emphasizes that DPSGD has great benefits, from the perspective of actual performance, the results are not very satisfactory when the learning rate is very large. Therefore, whether a large learning rate brings a great improvement in efficiency will be a very important question, and I don't think the author provide a detailed analysis in the text.

---

> ### Author Response · Authors · 2022-08-01
> **Response to  Reviewer 2WDQ (2/2)**
>
> Questions
>
> (1) The orthogonal decomposition of noise in Equation (4) is based on the expectation of users. In other words, it only holds when the number of users is very large. However, the experiment shown in Figure 2 only performs with 5 learners. In this case, is the results of the experiment consistent with the theoretical analysis?
>
> The decomposition of noise in Eq. 4 by itself does not depend on the number of users. However, whether or not we can approximate the distribution of the gradients from a finite number of  learners by their mean and variance is a valid question. To this end, we tried with less and more learners (3 or 7) in MNIST and we did not find any difference in terms of the ability of DPSGD in helping convergence with the general trend of reduced effective learning rate in the beginning of training (shown in Fig. 2b) unchanged. In the final version of the paper, we will mention these experiments. The intuitive explanation for this is that besides averaging over different learners the overall learning dynamics also depends on integration over time (iteration), which provides another averaging process that makes the Gaussian approximation a good one.
>
> (2) In Table 1, the best performance comes from when lr=1x, and the performance drops significantly when lr is very large. What are the advantages of a large learning rate?
>
> Table 1 is measured for the *same* number of epochs. With a larger batch size , one can use more GPUs. For example, it takes batch size 256 ~16x more time to finish than using a batch size 4096 because the latter can run on 16x more GPUs. Please note that for more “modern” architecture of CV tasks (e.g., ResNet, ResNext and DenseNet), DPSGD does recover the baseline accuracy in the large batch setting as reported in Appendix E.6 (Figure 9 and Table 10) in the supplementary materials.  For the reasons that the listed CV tasks (AlexNet, VGG, VGG-BN) in Table 1 are difficult to train in large batch settings please also see response to Weakness (2).
>
> Limitations
>
> (1) Even though the authors claim that they describe the limitations of their work. However, I cannot an explicit description. I would like to ask the authors if they can show me where is the limitation during the rebuttal.
>
> One limitation is that we need to introduce the  the assumption that $\delta\_i w|\mathcal{F}\_{t-1}$, which is Gaussian is a bit strong. However it gives us the cleanest possible way to illustrate that DPSGD is optimizing a smoother landscape (we show the smoothness constant in DPSGD case is smaller compared with SSGD). Without assuming this, the challenge is that we cannot exactly calculate the smoothness constant since there is no closed form.

---

> > ### Comment · Reviewer_2WDQ · 2022-08-09
> > **Update**
> >
> > Thank you for your detailed responses. After reading the rebuttals, I tend to keep my score.

---

> ### Author Response · Authors · 2022-08-01
> **Response to Reviewer 2WDQ (1/2)**
>
> Weakness
>
> (1) The writing of this work has a big problem. The authors put important context including Related Works and DPSGD and SSGD Runtime Comparison in the appendix. Related work is almost an indispensable chapter of an article. Besides, whether DPSGD has an advantage in runtime compared to SSGD is also important. Since the authors claim that DPSGD can have an advantage in terms of convergence, it is to be expected whether this advantage can bring about a saving in runtime. Although the conference has body content restrictions, putting very important content in the appendix is ​​a discouraged behavior.
>
> Related works are in Appendix G is the full paper (supplementary materials) and Runtime results are in Appendix F in the full paper (supplementary materials). In the final version, we will add the related works and runtime results in the main paper.
>
> (2) Although this article emphasizes that DPSGD has great benefits, from the perspective of actual performance, the results are not very satisfactory when the learning rate is very large. Therefore, whether a large learning rate brings a great improvement in efficiency will be a very important question, and I don't think the author provide a detailed analysis in the text.
>
> First, there is not a known general solution to maintain model accuracy (within the *same* number of training epochs) when increasing batch size [12,24,18,54,57,61]. One key research area of distributed training is how to reduce the model accuracy gap between small-batch baseline and large-batch distributed training setting. However, please note that DPSGD is able to recover the baseline accuracy in the large batch setting (or shorter warmup window) for ASR and NLP tasks.
>
> In the CV cases, as we discussed at the beginning of Section 4.1: On ImageNet-1K we test 6 CNN models – AlexNet, VGG11, VGG11-BN, ResNet-50, ResNext-50  and DenseNet-161. Among them, AlexNet and VGG have rougher loss landscapes and can only  work with smaller learning rates, while VGG11-BN, ResNet-50, ResNext-50, and DenseNet-161 have smoother loss landscapes thanks to the use of BatchNorm or Residual Connections, and thus  can work with larger learning rates. Table 1 listed in the main paper shows the most difficult large-batch CV tasks. However, we do show that DPSGD outperforms SSGD significantly in these tasks. Please note that for easier large-batch CV tasks (e.g., ResNet, ResNext and DenseNet) , DPSGD does recover the baseline accuracy in the large batch setting as reported in Appendix E.6 (Figure 9 and Table 10) in the supplementary materials.
>
> Second, since the SSGD is the de facto  distributed training approach. It is a common practice to increase learning rate to compensate for the reduced number of parameter updates in SSGD. The goal of this paper is to compare SSGD and DPSGD in such settings. The general tradeoff of training time benefits  and the model quality degradation caused by distributed training (SSGD , DPSGD or otherwise) is beyond the scope of this paper.

---

### Official Review · Reviewer_ndjb · 2022-07-09

**Rating:** 3
**Confidence:** 3
**Soundness:** 1 poor
**Presentation:** 1 poor
**Contribution:** 2 fair

**Summary:**

The paper studies two distributed learning algorithm for training large-scale deep learning models.
The authors argue that the decentralized parallel stochastic gradient descent (DPSGD) has a series of advantages over the more commonly employed Synchronous SGD (SSGD). They argue that the optimization dynamics of DPSGD injects noise which smooths the optimization landscape and allows for larger learning rates. This is especially beneficial in the large-batch-size setting.
The paper contains one main theoretical result concerning the smoothness of the objective function that DPSGD implicitly minimizes.
Several experiments with various tasks and architecture are conducted.

**Questions:**

*Major*
- I do not understand why the assumption that $\delta_i w |\\mathcal{F}_{t-1}$ should be Gaussian with zero mean and especially equal (scalar) variance. Why this should make sense? In the appendix the authors call for the central limit theorem, but 1) I do not understand to what it should be applied 2) I think that at least the multidimensional version of the CLT should be considered here. On the contrary, the empirical results with SSGD + Gaussian noise (which show poor performances) seem to suggest that there is a fundamental difference between the noise induced by DPSGD and Gaussian noise.
- One might hypothesize that the advantage of  DPSGD is that the noise is adaptive (but still Gaussian). To empirically verify this, the authors could run an experiment where the Gaussian noise added to SSGD is scheduled after what the authors find by running DPSGD (e.g. figure 2 b) for the MNIST setting. If the resulting methods performs on par with DPSGD, then I could be more willing to accept the assumption, although questions would remain.
- Otherwise, I would argue that Th 1. is not useful to understand the advantages of DPSGD.

*Others*
- From Lemma 2 of Random Gradient-Free Minimization of Convex Functions [39] it would follow that the smoothness coefficient should be $\frac{\sqrt{n}}{\sigma} G$ where $n$ is the dimension of the parameter vector. From where do you obtain $\frac{2}{\sigma} G$ that you report in the paper?
- I do not quite understand how the theoretical result cannot involve the mixing matrix. As the authors note, for a complete mixing matrix DPSGD is equivalent to SSGD; this should be reflected in the statement and formulation of the theorem.
- In eq. (2) is it intentional that there are 2 sets of weights $w_j$ and $w_{s,j}$? Isn't it that the loss is also computed at $w_{s,j}$? Or does each worker maintain 2 sets of weights?

*Minor*
- Why only limiting to the cross-entropy loss?
- I do not understand the reason behind "self" of self-adjusting. Why "self"? Usually one speaks about adaptive method.
- Typos: allow [for]; tasks -> models in line 29; inconsistencies between figures and tables and texts (e.g. Fig 1 SSGD-noise vs SSGD* in the text).


**Limitations:**

- Negative societal impacts are not discussed, but I do not see any visible one, so I think missing that discussion is fine for this paper.
- Limitations of the proposed analysis are not discussed, and should be added, e.g. in relation to the assumptions of the theorem.
- I do not think it is proper to have one paragraph sketching out very informally a possible result suggesting that other "people" (line 144) then can fill the gaps. One might suggest this as a potential interesting future work (briefly and in the conclusions). Or spell out the theorem and include the proof. As it stands, the passage starting at 144 should be removed.

**Strengths And Weaknesses:**

**Strengths:**
- The experimental section contains several diverse tasks and architectures, therefore having large coverage
- The intuitive and empirical arguments in favor of DPSGD are interesting; somewhat helping build up confidence in this algorithm which has other advantages wrt SSGD

**Weaknesses:**
- I am not quite convinced of the utility of the theoretical results, especially regarding the Gaussian hypothesis (see below in questions)
- Even factoring this concern out, I believe the theoretical contribution to be very limited as it is basically the application of a know Lemma
- The presentation is unclear in various passages, including the proof of the theorem; notation is too often overloaded (see eq (3)) making it difficult to understand what is what. This is especially bad in section 2.2: e.g. in equation (4) one might expect the authors to discuss about the DPSGD algorithm, but there is no mention of the gradient $g_j=\nabla L^{\mu_j}(w_j)$. Other important quantities such as the total loss are never explicitly defined and the reader is left with the need to self-interpret the paper at various points.
- The layout of the paper should be revised: the abstract is too long and the related work section cannot be relegated to the appendix. Space can be saved by avoiding repetitions, focusing discussion and removing unnecessary "summary" sentences at the end of each experimental section.
- The claim that DPSGD needs very little to no tuning is exaggerated and not backed up by the experiments. In particular, note that in the NLP experiment DPSGD still needs the learning rate warm-up stage, although shorter.
- The paper would benefit from a more complete (also formally) introduction of the two algorithms. For instance it is not clear what happens at the end of the optimization with DPSGD. Are the weights averaged over all the worker?

---

> ### Author Response · Authors · 2022-08-01
> **Response to Reviewer ndjb (3/3)**
>
> Limitations:
>
> (1) Limitations of the proposed analysis are not discussed, and should be added, e.g. in relation to the assumptions of the theorem.
>
> We need to assume that the noise in the weight is independent Gaussian noise, which may not be realistic. However, assuming it can help us understand the effect of landscape smoothing effect.
>
> (2) I do not think it is proper to have one paragraph sketching out very informally a possible result suggesting that other "people" (line 144) then can fill the gaps. One might suggest this as a potential interesting future work (briefly and in the conclusions). Or spell out the theorem and include the proof. As it stands, the passage starting at 144 should be removed.
>
> Thank you for your suggestion. We will put it in the conclusion section to suggest future works.

---

> > ### Comment · Reviewer_ndjb · 2022-08-07
> > **Reply**
> >
> > I thank the authors for their reply, which satisfies some of my concerns, but unfortunately not the major ones.
> > I believe the theoretical results of the paper needs to be revised, to establish a much clearer link between theory and proposed algorithm. Furthermore the authors have not answered satisfactorily to Q2.
> > Overall the algorithm presented and the intuition behind it seem to be rather interesting, however I do not think that the paper in its current form is ready for publication. I will keep my original score.

---

> ### Author Response · Authors · 2022-08-01
> **Response to Reviewer ndjb (2/3)**
>
> Questions:
>
> Major:
>
> (1) I do not understand why the assumption that $\delta\_i w |\mathcal{F}\_{t-1}$ should be Gaussian with zero mean and especially equal (scalar) variance. Why this should make sense? In the appendix the authors call for the central limit theorem, but 1) I do not understand to what it should be applied 2) I think that at least the multidimensional version of the CLT should be considered here. On the contrary, the empirical results with SSGD + Gaussian noise (which show poor performances) seem to suggest that there is a fundamental difference between the noise induced by DPSGD and Gaussian noise.
>
> $\delta_i w|\mathcal{F}\_{t-1}$ stands for the noise in the weight, while the SSGD+Gaussian noise means that the noise is added on gradient. They are two different types of noise. The first type of noise is inherent noise due to the DPSGD algorithm which helps smoothen the loss landscape, while the second type of noise is artificial Gaussian noise.
>
> We acknowledge that the assumption that $\delta\_i w|\mathcal{F}\_{t-1}$ is Gaussian is a bit strong. However it gives us the cleanest possible way to illustrate that DPSGD is optimizing a smoother landscape (we show the smoothness constant in DPSGD case is smaller compared with SSGD). Without assuming this, we cannot exactly calculate the smoothness constant since there is no closed form. Even though Gaussian noise  is a simplified assumption, it is a useful model to interpret the optimization dynamics.
>
> (2) One might hypothesize that the advantage of DPSGD is that the noise is adaptive (but still Gaussian). To empirically verify this, the authors could run an experiment where the Gaussian noise added to SSGD is scheduled after what the authors find by running DPSGD (e.g. figure 2 b) for the MNIST setting. If the resulting methods performs on par with DPSGD, then I could be more willing to accept the assumption, although questions would remain. Otherwise, I would argue that Th 1. is not useful to understand the advantages of DPSGD.
>
> Our results are consistent with the reviewer’s hypothesis that DPSGD effectively introduces an adaptive learning rate that depends on the loss landscape – the rougher the landscape the smaller the effective learning rate, which allows the algorithm to converge even with large “bare” learning rate. As shown in Fig. 4 in the Appendix, in the initial stage of learning, the strength of the DPSGD specific noise $\Delta^{(2)}$  is much larger (by 1-2 orders of magnitude) than the SSGD noise. As the system learns and the landscape becomes smoother,  $\Delta^{(2)}$ decreases and it reaches the same level as the SSGD noise after the system passes the initial learning stage and stays constant (with large fluctuations) afterwards. Therefore, adding a Gaussian noise with a constant strength after the system passes through the initial rough landscape should have similar performance as DPSGD noise, which is only critical for convergence in the initial stage of learning.
>
> Others:
>
> (1) From Lemma 2 of Random Gradient-Free Minimization of Convex Functions [39], it would follow that the smoothness coefficient should be $\frac{\sqrt{n}}{\sigma}G$, where $n$ is the dimension of the parameter vector. From where do you obtain $\frac{2}{\sigma} G$ that you report in the paper ?
>
> Sorry we missed a $\sqrt{n}$. We will add them in the revision.
>
> (2) I do not quite understand how the theoretical result cannot involve the mixing matrix. As the authors note, for a complete mixing matrix DPSGD is equivalent to SSGD; this should be reflected in the statement and formulation of the theorem.
>
> The theoretical result requires that every machine has a different weight. If the difference between the weight and the averaged weight satisfies the i.i.d. Gaussian, then we can show that DPSGD is doing optimization on a more smooth landscape. We will add more details in the revision.
>
> (3) In eq. (2) is it intentional that there are 2 sets of weights $w_j$ and $w_{s,j}$ ? Isn’t it that the loss is also computed at $w_{s,j}$ ? Or does each worker maintain 2 sets of weights ?
>
> No, gradients are calculated w.r.t $w_j$. Each worker maintains 2 sets of weights so that gradient calculation and weights communication can happen concurrently.
>
> Minor:
>
> (1) Why only limiting to the cross-entropy loss?
>
> It is the most commonly used loss function in these CV, NLP and ASR tasks.
>
> (2) I do not understand the reason behind "self" of self-adjusting. Why "self"? Usually one speaks about adaptive method.
>
> Noise strength changes automatically according to the loss landscape (no manual tuning is needed) in DPSGD. DPSGD doesn’t introduce any new hyper-parameter to tune, the loss-landscape dependent noise effect comes for free (i.e., inherent system noise).
>
> (3) Typos: allow [for]; tasks -> models in line 29; inconsistencies between figures and tables and texts (e.g. Fig 1 SSGD-noise vs SSGD* in the text).
>
> Thanks! We will fix these.

---

> ### Author Response · Authors · 2022-08-01
> **Response to Reviewer ndjb (1/3)**
>
>
> Weakness:
>
> (1) I am not quite convinced of the utility of the theoretical results, especially regarding the Gaussian hypothesis (see below in questions)
>
> (2) Even factoring this concern out, I believe the theoretical contribution to be very limited as it is basically the application of a know Lemma
>
> Gaussian noise helps us better understand and analyze why the noise can make the landscape smoother. Considering general noise would make the analysis extremely difficult, and we plan to study this problem in the future work.
>
> (3) The presentation is unclear in various passages, including the proof of the theorem; notation is too often overloaded (see eq (3)) making it difficult to understand what is what. This is especially bad in section 2.2: e.g. in equation (4) one might expect the authors to discuss about the DPSGD algorithm, but there is no mention of the gradient $g\_j=\nabla L^{\mu\_j}(w\_j)$. Other important quantities such as the total loss are never explicitly defined and the reader is left with the need to self-interpret the paper at various points.
>
> We apologize for the complexity of notations in this paper, which we will try to simplify and clarify in the final version of the paper. $g\_j=\nabla L^{\mu\_j}(w\_j)$ represents the minibatch gradient for learner-j at weight $w_j$. Since we are interested in the learning dynamics of the whole system including all learners, we focused on the statistical properties of $g_j$ over all learners in particular the average gradient $g_a$ over all learners and their variance ($\Delta\_{DP}$, $\Delta\_S$, $\Delta^{(2)}$) rather than gradient from individual learners. The total loss for a given learner is the average of its minibatch losses, which is taken to be the cross-entropy loss as described in line-93 of our paper.
>
> (4) The layout of the paper should be revised: the abstract is too long and the related work section cannot be relegated to the appendix. Space can be saved by avoiding repetitions, focusing discussion and removing unnecessary "summary" sentences at the end of each experimental section.
>
> Thank you for the suggestions! The related works are in Appendix G of the full paper (i.e., supplementary materials). In the final version of the paper, we will include the related works section.
>
> (5) The claim that DPSGD needs very little to no tuning is exaggerated and not backed up by the experiments. In particular, note that in the NLP experiment DPSGD still needs the learning rate warm-up stage, although shorter.
>
> We didn’t claim DPSGD can automatically pick up the optimal hyper-parameter setup (we are unaware of any work that claims to do that), we claim that DPGSD allows more freedom to choose hyper-parameter from (i.e, larger range of learning rate) than SSGD. Please note that DPSGD doesn’t introduce any new hyper-parameter, it simply uses whatever hyper-parameter practitioners might use in SSGD, which is the de-facto distributed training algorithm.
>
> (6) The paper would benefit from a more complete (also formally) introduction of the two algorithms. For instance it is not clear what happens at the end of the optimization with DPSGD. Are the weights averaged over all the worker?
>
> We presented the introduction to these two algorithms at the beginning of Section 2.
>
> For the large-scale experiments, as mentioned in Section 4, the implementation is based on [33], with  a randomized  mixing matrix as in [59].  By the end of optimization, the weights are averaged across learners -- in practice, the model quality is no different (of any significance) than picking one model from any learner.

---

### Official Review · Reviewer_x6PL · 2022-07-13

**Rating:** 3
**Confidence:** 3
**Soundness:** 2 fair
**Presentation:** 2 fair
**Contribution:** 2 fair

**Summary:**

The authors propose a comparison between two methods of parallelization of the SGD, namely SSGD and DPSGD. They prove theoretically that the decentralized one (DPSGD) adds a smoothing effect to the naive SSGD, and shows that DPSGD is better than SSGD as the batch size increases.

**Questions:**

There is a beginning of theoretical analysis of the proposed method though the Hessian of the loss. SSGD and DPSGD provide information about the loss surface, and, intuitively, DPSGD provides richer information. Is it possible to formulate SSGD and DPSGD as second-order methods, where the approximation of the preconditioning matrix (usually the Hessian) is computed in a distributed way? Is this the optimal second-order method with acceleration through distributed computing? Etc.

Note: Fig2b shows an evolution of the learning rate, which appears also when using second-order methods.

**Limitations:**

The authors have tested an improved version of SSGD against DPSGD, in order to compare them fairly.

The main unchecked limitations are (as mentioned above):
 * no error margins in the results;
 * lack of reproducibility: the reader lacks information to obtain the same hyperparameters.

**Strengths And Weaknesses:**

## Strengths

This paper makes a tiny link between some distributed methods and second-order methods. It helps to understand their dynamics. Furthermore, the computation of an "effective learning rate" for DPSGD is useful to illustrate the implicit mechanisms of this method.

The experiments on large datasets, such as ImageNet, are valuable.

According to the experiments, DPSGD seems to perform way better than the alternative choices when the batch size increases.

### Control experiments

It is noticeable that the authors have tested a modification of the SSGD, called ``SSGD*'', in order to test fairly SSGD against DPSGD.

## Weaknesses

### Theory

Theorem 1 contains at least two major issues:
 * The hypothesis: ``$\| \frac{1}{n} \sum\_{i = 1}^n X\_i - \frac{1}{n-1} \sum\_{i = 1}^{n - 1} X\_i \| \leq \epsilon$ almost surely'' is not reasonable when the $X\_i$ are i.i.d. and without compact support. So, we cannot consider the simple case where the $X\_i = \nabla L^{\mu\_i(t)}(\vec{w}\_i(t))$ are i.i.d. Gaussian random variables.
 * In the proof: it is true that, conditionally to $\mathcal{F}\_{t-1}$, the $\vec{w}\_{i}(t)$ are independent, but it is false for the $\delta w\_i(t)$.
Let us recall that: $\delta \vec{w}\_{i}(t) = \vec{w}\_{a}(t) - \vec{w}\_{i}(t) = \frac{1}{n} \sum\_{j = 1}^n \vec{w}\_{j}(t) - \vec{w}\_{i}(t)$.Thus, all $\delta \vec{w}\_{i}(t)$ are linear combinations of the same set of variables, with only non-zero coefficients. It is thus impossible that two $\vec{w}\_{i}(t)$ are independent. Besides, independence has nothing to do with the fact the $\sum_{i=1}^n \delta \vec{w}\_{i}(t) = 0$... on the contrary, this equation indicates a strong dependence between the terms.

However, it is probable that this theorem can be made useful and rigorous. But, anyway, it does not seem to be useful in the presented work.

### Experiments

Error margins: there is no mention of any variance or error margin in the results. So, we cannot evaluate the significance of the presented results: if a reported result seems to be better than another, it may be luck...

The main drawback of the experimental setup is its complexity. In particular, a huge preliminary phase of fine-tuning seems to be necessary, without proposing a reproducible procedure to recover the proposed hyperparameters (which are very numerous). The authors only provide citations of preceding works, where the fine-tuning has already been done.

It is clear that the authors claim to prove the reliability of DPSGD as the batch size increases; they do not claim to beat the state of the art. So, there is no point of testing architecture trained with highly fine-tuned optimization procedures. A unified and simplified optimization process should have been considered.

### Clarity

The paper suffers from several writing issues.
 * The "related works" section must not be put in appendix. This section is fundamental to draw a line between existing works and actual contributions, and to evaluate fairly the significance of the paper.
 * Using $\vec{w}_j$, where $j$ is an index standing for an integer, and $\vec{w}_a$, where $a$ is a simple letter (standing for "average"), is confusing. Why not using $\hat{w}$ instead of $\vec{w}_a$? Same issue with $\vec{w}_{s, j}$. Besides, arrows can easily be removed, or replaced by bold text: $\mathbf{w}_j$ instead of $\vec{w}_j$.
 * If the authors do not use the general version of DPSGD presented in [33], then the general notation $\vec{w}_{s, j}$ can be removed.

---

> ### Author Response · Authors · 2022-08-01
> **Response to Reviewer x6PL (3/3)**
>
> Questions
>
> (1) There is a beginning of theoretical analysis of the proposed method though the Hessian of the loss. SSGD and DPSGD provide information about the loss surface, and, intuitively, DPSGD provides richer information. Is it possible to formulate SSGD and DPSGD as second-order methods, where the approximation of the preconditioning matrix (usually the Hessian) is computed in a distributed way? Is this the optimal second-order method with acceleration through distributed computing? Etc.
>
> Note: Fig2b shows an evolution of the learning rate, which appears also when using second-order methods.
> The reviewer is correct that we are interested in theoretical analysis (not in the form of theorem proving but rather a physics style understanding) of both algorithms (SSGD and DPSGD) in terms of the loss landscape characterized in part by the Hessian of the loss. As shown in previous work (e.g., ref. 8), all SGD based algorithms introduce a landscape-dependent (Hessian-dependent) noise that has a similar effect as a second-order method without computing the Hessian explicitly, which is computationally expensive.
> As we explained in the paper (line 171-179), even though SSGD has a Hessian-dependent noise, its amplitude is proportional to 1/(nB), which is small given a large number of learners (n). For DPSGD, in addition to the SSGD noise, there is another noise term $\Delta^{(2)}$ caused by the asynchrony of the learners. As shown in Eq.5, this additional noise term also depends on the Hessian. As we stated in our paper (line 177-179), “A main finding of our paper is that the additional landscape-dependent noise $\Delta ^{(2)}$ in DPSGD can make up for the small SSGD noise when nB is large and help enhance convergence in the large batch setting.”
> Since the additional landscape-dependent noise $\Delta^{(2)}$ originates from the difference in weights among the different learners (C in Eq. 5 is the covariance matrix of weights from different learners), as the reviewer may have guessed, DPSGD (but not SSGD) represents a distributed way of computing landscape information (e.g., Hessian).
> Finally, we thank the reviewer for pointing out the similarity between the evolution of the effective learning rate (shown in Fig. 2b) and that from the second-order methods, which we will mention in the final version of the paper.
>
>
> Limitations
>
> (1) no error margins in the results
>
> Please see response to Experiments (1).
>
> (2) lack of reproducibility: the reader lacks information to obtain the same hyperparameters
>
> In Section 4.1 (Computer Vision), 4.2 (Automatic Speech Recognition), and 4.5 (Natural Language Processing),  we clearly stated the exact hyper-parameter setup (batch-size, learning rate, warmup schedule) one needs to reproduce the results.

---

> > ### Comment · Reviewer_x6PL · 2022-08-08
> > **Post-rebuttal**
> >
> > I acknowledge the authors' response.
> > I maintain my rating, since there is no convincing theoretical argument in favor of the proposed method. This is a major problem, since the authors claim to provide such arguments.
> > Either the authors give up the theoretical analysis, either they make it sound. Anyway, it cannot be achieved in a minor revision of the paper.

---

> ### Author Response · Authors · 2022-08-01
> **Response to Reviewer x6PL (2/3)**
>
>
> Experiments
>
> (3) It is clear that the authors claim to prove the reliability of DPSGD as the batch size increases; they do not claim to beat the state of the art. So, there is no point of testing architecture trained with highly fine-tuned optimization procedures. A unified and simplified optimization process should have been considered.
>
> One key challenge of distributed training is how to tune hyper-parameters when batch size increases -- there is no known general solution. Some common practices include learning rate warmup and linear scaling . We show in such practice, DPSGD can work with a larger range of hyper-parameters than SSGD, which is the de-facto distributed training algorithm.
>
> It is a common practice to use publicly known best hyper-parameters from the research literature when comparing different training algorithms. It is common knowledge among practitioners that different training tasks  require different hyper-parameter setup (learning rate, batch size and etc).  We are unaware of any work when evaluating across different domains adopts the same set of hyper-parameters or a “unified and simplified optimization process”.
>
> Finally, it is hard to only prove the reliability of DPSGD without showing competitive performance compared to SOTA in order to publish.  Almost all the existing high performance deep learning models are after heavy hyperparameters optimization depending on the task of interest and architecture under investigation.  In order to compare with published results and at least achieve competitive results, it is a reasonable strategy to start DPSGD with a similar optimization process as the published ones.
>
> Clarity
>
> (1) The "related works" section must not be put in appendix. This section is fundamental to draw a line between existing works and actual contributions, and to evaluate fairly the significance of the paper.
>
> The related works are in Appendix G of the full paper (i.e., supplementary materials). We will address this in the final version of the paper.
>
> (2) Using $\vec{w}\_j$, where $j$ is an index standing for an integer, and $\vec{w}\_a$, where $a$ is a simple letter (standing for "average"), is confusing. Why not using $\hat{w}$ instead of $\vec{w}\_a$ ? Same issue with $\vec{w}_{s,j}$. Besides, arrows can easily be removed, or replaced by bold text: $\mathbf{w}\_j$ instead of $\vec{w}\_j$.
>
> Thanks for the suggestion. To avoid confusion, we will change the notation to use $\hat{w}$ for the average weights over all learners and $\hat{w}\_j$ for the averaged weights for learner-j in the final version of the paper.
>
> (3) If the authors do not use the general version of DPSGD presented in [33],  then the general notation $\vec{w}\_{s, j}$ can be removed
>
> $\vec{w}\_{s,j}(t)$ represents the locally averaged weight (at iteration t) over a subset of learners that a given learner-j picks according to the mixing matrix. Learner-j uses $\vec{w}\_{s,j}(t)$ as its starting weight for its weight update for the next iteration (t+1) as shown in Eq. 2.
>
> For the large-scale experiments conducted in this paper, as mentioned in the first paragraph of Section 4, each DPSGD learner picks a random neighbor in each iteration as done in [59]. The theoretical analysis in this paper assumes a general version of the mixing matrix presented in [33].

---

> ### Author Response · Authors · 2022-08-01
> **Response to Reviewer x6PL (1/3)**
>
> Weakness
>
> Theory
>
> (1) The hypothesis: ``$\frac{1}{n} \sum_{i = 1}^n X_i - \frac{1}{n-1} \sum_{i = 1}^{n-1} X_i \leq \epsilon$ almost surely'' is not reasonable when the $X_i$ i.i.d. and without compact support. So, we cannot consider the simple case where the $X_i$ = $\nabla L^{\mu_i(t)}$
> ($\vec{w}_i(t)$) are i.i.d. Gaussian random variables.
>
> Thank you for pointing it out. We can relax the requirement of “almost surely” to be high probability. This is true when $n$ is sufficiently large as mentioned in the statement of Theorem 1. The reason is that the variance gets reduced when $n$ is large and both $\frac{1}{n} \sum_\{i = 1}^n X_i$ and $\frac{1}{n-1} \sum_{i = 1}^{n-1} X_i$ concentrate around the same mean with very small deviation with high probability. Then the whole theorem holds with high probability. We will make it more clear in the revision.
>
> (2) In the proof: it is true that, conditionally to $\mathcal{F}\_{t-1}$ , the $\vec{w}\_i(t)$ are independent, but it is false for the $\delta w_i(t)$.
> Let us recall that: $\delta \vec{w}\_{i}(t) $ = $\vec{w}_a(t) - \vec{w}_i(t)$ = $\frac{1}{n}$ $\sum\_{j=1}^{n}$  $\vec{w}\_{j}(t) - \vec{w}\_{i}(t)$. Thus, all $\delta \vec{w}\_{i}(t)$ are linear combinations of the same set of variables, with only non-zero coefficients. It is thus impossible that  two $\vec{w}\_{i}(t)$ are independent. Besides, independence has nothing to do with the fact the $\sum\_{i=1}^n \delta \vec{w}\_{i}(t) = 0$. On the contrary, this equation indicates a strong dependence between the terms.
>
> Thank you for your comments. We agree with this. The reason for assuming the i.i.d. $\delta w\_i(t)$ is that it provides a cleanest possible approach for supporting our argument: DPSGD is optimizing a smoother function. We can add a remark on this assumption in the revision and acknowledge that we cannot prove it rigorously. In addition, we can add a paragraph illustrating the intuitive reason why it smoothen the landscape by assuming i.i.d. Gaussian $\delta w\_i(t)$.
>
> Experiments
>
> (1) Error margins: there is no mention of any variance or error margin in the results. So, we cannot evaluate the significance of the presented results: if a reported result seems to be better than another, it may be luck...
>
> First, please note that In large-scale NLP, ASR experiments like ours, it is customary that no variance is reported. See Transformer[50], BERT[6], GPT[42,43], and various ASR papers [15,57].  These experiments are expensive enough that the common practice is not to do multiple runs.
>
> Second, for all the numbers we reported in the experimental results section of the paper, we keep two decimal points. For the SSGD vs DPSGD numbers reported in Table 1, 2 and 5 large-scale runs (CV, ASR,NLP), we ran each experiment at least 3 times. The variance of heldout-loss for ASR and NLP tasks are effectively 0.00 (two decimal points). The variance of test accuracy for CV tasks is consistently smaller than 0.10. The difference between SSGD and DPSGD is several orders of magnitude higher (e.g., a held loss 10.37 vs 1.47 in ASR tasks).  We can make a remark about variance in our final version.
>
> Third, we grid search over learning rates and gaussian noise strength  as demonstrated in Section 4.3 to show that DPSGD consistently outperforms SSGD in the large batch setting.
>
> Finally, we show an advantage for DPSGD across three different tasks using different architectures is a strong indication that we aren’t just getting luck with one or two experiments.
>
> (2) The main drawback of the experimental setup is its complexity. In particular, a huge preliminary phase of fine-tuning seems to be necessary, without proposing a reproducible procedure to recover the proposed hyperparameters (which are very numerous). The authors only provide citations of preceding works, where the fine-tuning has already been done.
>
> First, we didn’t claim DPSGD can automatically pick  the optimal hyper-parameter setup, we claim that DPGSD gives practitioners more freedom to choose hyper-parameter from (i.e., a larger range of learning rate).
>
> Second, all the hyper-parameter setup we use are from the research literature that use SSGD and therefore such hyper-parameter setups are optimized for the SSGD. We use them “as is” for DPSGD. In another word, those hyper-parameters are “fine-tuned” for SSGD, but not for DPSGD.

---

### Meta-Review · Area_Chair_aiBS · 2022-08-26

**Recommendation:** Reject
**Confidence:** Certain

**Metareview:**

This paper compares all-reduce SGD (SSGD) with decentralized SGD (DPSGD) and argues that the latter can tolerate lager stepsize due to a smoothing effect induced by noise in DPSGD.

The reviewers found that the theoretical contribution is overclaimed. By the strong assumptions needed in the theory section (such as e.g. assuming Gaussian updates) the analysis becomes somewhat disconnected from the experiments, and, in addition, reviewers found several typos and issues in Section 2 of the original submission. Even though the numerical evaluation was judged more positively by all reviewers (and championed by one), we came to the consensus that the paper should be rejected in its current form.

(Minor comments:) In the discussion, we also found that that the term “self-adjusting” might be a bit misleading (as learning rates are kept fixed and are not self-adjusting), and that the paper would benefit of a brief discussion of related works that study the benefitting effect of smoothing in large-batch training (such as https://arxiv.org/abs/1805.07898 or https://arxiv.org/abs/1906.10822, etc.).


**Award:**

No

---

### Decision · Program_Chairs · 2022-09-14

Reject